

# Marine and Continental Stratocumulus Cloud Microphysical Properties Obtained from Routine ARM Cimel Sunphotometer Observations

Kaiden Sookdar[1], Scott E. Giangrande[2], John D. Rausch[2], Lihong Ma[2], Meng Wang[2], Dié Wang[2], Michael
P. Jensen[2], Ching-Shu Hung[3], and J. Christine Chiu[3]

[1]Earth and Atmospheric Sciences Department, Cornell University, Ithaca, NY
[2]Environmental and Climate Sciences Department, Brookhaven National Laboratory, Upton, NY, USA
[3]Department of Atmospheric Science, Colorado State University, Fort Collins, CO, 80523, USA

*Correspondence to*: Scott E. Giangrande (sgrande@bnl.gov)

**Abstract.** This study investigates marine and continental stratocumulus (Sc) cloud properties obtained from an automated implementation of a multispectral photometer retrieval. Photometer methods simultaneously retrieve cloud optical depth ($\tau$) and cloud droplet effective radius ($r_e$), with estimates for liquid water path (LWP) calculated on the availability of those quantities. These efforts evaluate retrieved cloud properties for Sc identified during a recent 6-year period collected over the U.S. Department of Energy Atmospheric Radiation Measurement (ARM) program sites in Oklahoma, USA (SGP) and in the

Azores, Portugal (ENA).

Modest agreement in key quantity retrievals is found between the routine photometer products and multisensor collocated profiling references. Cumulative breakdowns contingent on cloud thickness indicate increases in all retrieved quantities in thicker clouds, with larger discrepancies in the relative performance between the retrievals collected in the presence of drizzle. Under continental cloud conditions, the clouds of a similar thickness and $r_e$ to those sampled under marine

conditions report a factor of 1.5 larger $\tau$ and LWP. An $R^2 \cong 0.65$ is found between photometer $\tau$ retrievals and shadowband radiometer measurements, with photometer retrievals reporting a high (relative) bias. The $\tau$ intercomparisons indicate that variability between retrievals is a factor of three larger than errors reported from individual retrieval input perturbation tests. Photometer $r_e$ retrievals suggest a low $R^2$ ($< 0.1$) having a standard deviation $\cong 3$ μm when compared to ARM baseline multi-sensor radar/radiometer references (accounting for offsets in the cloud droplet number concentration assumptions of the latter).

However, photometer LWP calculations remain relatively unbiased in non-drizzling conditions, with errors $O[50$ g m$^{-2}]$ and $R^2 \cong 0.5$ to collocated radiometer and interferometer references. Additional sensitivity tests for island influences on ENA marine Sc properties suggest that while oceanic versus island-influenced winds may promote significant shifts in quantities, this influence is lower than retrieval method uncertainty and/or collocated instrument variability.



## 1 Introduction

A primary source of uncertainty in Earth system model (ESM) predictions of climate change is in the representation of cloud processes and associated cloud feedback (e.g., IPCC, 2013). Several fundamental cloud properties critical to the understanding of aerosol-cloud interactions are poorly constrained by observations, with key deficiencies in our observations of cloud and precipitation droplet sizes and cloud optical depth. Observations of these cloud properties are often challenging to estimate from remote-sensing platforms and costly to obtain from *in situ* aircraft – requiring extensive instrument calibration,

conditioning, and computational methods to retrieve the desired quantities. Nevertheless, advancing cloud observations and techniques is critical to an improved understanding of cloud formation, dissipation, aerosol-drizzle interaction, radiative impacts, and related model process studies (e.g., Albrecht, 1989; McComiskey et al., 2009). The observations of boundary layer clouds, and improved knowledge of stratocumulus cloud (Sc) processes and properties, are especially important to this ESM advancement. This is because these clouds have extensive coverage and exert controls on boundary layer dynamics and

the global radiative energy balance (e.g., Klein, 1997; Bony & Dufresne, 2005; Hartmann et al., 1992; Klein & Hartmann, 1993; Wood et al., 2015; Sherwood et al., 2020). Understanding capabilities for retrieving Sc cloud properties is essential, as small changes in Sc coverage, thickness and cloud droplet properties can impart significant net radiative changes (e.g., Hartmann et al., 1992; Wood et al., 2012).

It is uncommon that observations simultaneously provide information on the cloud droplet sizes, cloud optical depth, and/or liquid water path (LWP). One such capable instrument is a multispectral photometer. The U.S. Department of Energy (DOE) Atmospheric Radiation Measurement (ARM) user facility has deployed photometers at its fixed and mobile facility deployments for over two decades (e.g., Mather and Voyles, 2013; Miller et al., 2016). As a narrow field of view (FOV, 1.2°) instrument, one advantage of this instrument is in its viability for sampling a range of broken to overcast cloud cover conditions.

Originally designed to retrieve aerosol optical properties, it was suggested by Marshak et al. (2004) and later expanded by Chiu et al. (2006; 2010; 2012) that the NASA AERosol RObotic NETwork (AERONET; Holben et al., 1998) implement a "cloud mode" strategy for its multispectral photometers (ARM's Sun-Sky-Lunar Multispectral Photometer). This mode is performed using two-channel radiance measurements during instrument sequences where clouds completely block the sun. When operated in this fashion, the mode enables estimates of the cloud optical depth ($\tau$). Recently, ARM upgraded its

photometers to a three-channel (440, 870, and 1640 nm wavelength) configuration to further constrain retrievals that simultaneously capture $\tau$ and cloud particle effective radius (Chiu et al., 2012). While previous two-channel (440, 870 nm) methods were applicable over vegetated land surfaces, this third channel constraint enables retrievals over ocean and ice surfaces, suitable for a range of higher-latitude and shipborne deployments (e.g., Wood et al., 2015; Lubin et al., 2020; McFarquhar et al., 2020; Wang et al., 2022; Geerts et al., 2022).






A motivation for this study is to evaluate advancements in cloud products derived from photometer measurements that are now being implemented on a routine basis in support of cloud process studies and validation for satellite, aircraft, and related retrieval applications (e.g., Minnis et al., 2011; Zhao et al., 2012; Bennartz and Rausch, 2017; Yang et al. 2018; Zhu et al., 2022). Our efforts draw on the extended ARM measurement record with a goal to deliver photometer-retrieved quantities of cloud properties as a baseline, continuing operational product. Overcast warm boundary layer cloud conditions were targeted since we have greater confidence in retrievals of their properties as compared to mixed-phase, ice, or broken cloud conditions (Section 2). Stratocumulus conditions are common over ARM's Eastern North Atlantic (ENA) site, while overcast low clouds are also frequent at ARM's Southern Great Plains (SGP, Sisterson et al., 2016) site; these sites will serve as our initial testbeds. Performance is explored for the photometer cloud retrieval quantities – the $\tau$ and the cloud droplet effective radius ($r_e$) – and a LWP estimated from those quantities. Results, discussions and physical interpretations for these product comparisons are offered in Sections 3 and 4 for SGP and ENA, respectively. Our results compile observations drawn from a 6-year period at both sites spanning datasets collected in 2014 through 2019. Methods to track uncertainty (i.e., constrained input perturbations, wavelength-contingent variability in radiance measurements and surface albedo) have been used to partially address uncertainty quantification. We conclude with key outcomes from this study in Section 5.

## 2 Data and Methods

Datasets were collected by the U.S. DOE ARM user facility at its Southern Great Plains (SGP) site in Lamont, Oklahoma (36.607 N, 97.487 W), and its Eastern North Atlantic (ENA) facility at Graciosa Island (39.091 N, 28.025 W) in the Azores archipelago (e.g., Mather and Voyles, 2013). We consider datasets from SGP and ENA for a 6-year window (Section 2.2). To target Sc events, these data were filtered according to a simple classification procedure and the availability of collocated reference instruments (Section 2.1). All retrievals have been averaged to a common 5-minute sampling window, based on the collection sequence timing of the photometer. In Fig. 1, we provide an example time-height display for baseline cloud observations and photometer $\tau$ retrievals from a qualifying event over the ENA site.

### 2.1 Stratocumulus Cloud Designation and Dataset Climatology

Stratocumulus conditions were selected based on a series of dataset availability and cloud property checks. First, we identify "events" where collocated Multifilter Rotating Shadowband Radiometer (MFRSR) observations were available, as required for $\tau$ retrieval comparisons (e.g., Min and Harrison, 1996; Min et al., 2003). Since ARM does not produce its MFRSR $\tau$ retrievals unless there are locally overcast conditions (the product defines this as 90% cloud cover from an effective 160° FOV, inferred from downwelling shortwave irradiances, e.g., Long et al., 2006), MFRSR retrieval availability serves as an initial check for overcast conditions. These conditions should maximize agreement between photometer and MFRSR, as broken cloud conditions may increase three-dimensional cloud-heterogeneity ambiguities on these retrievals (e.g., Turner et al., 2004; Masuda et al., 2019).



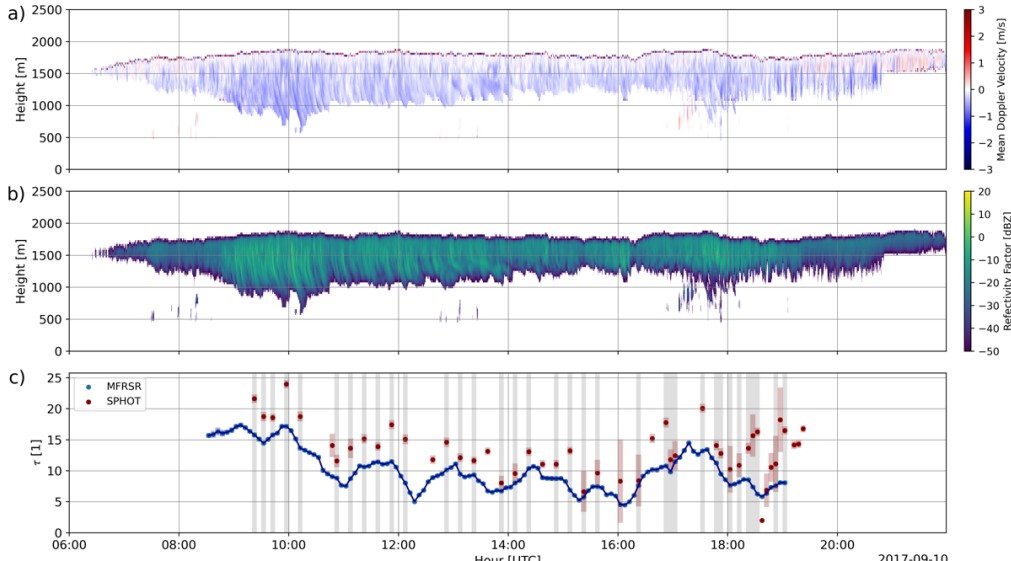

**Figure 1: The KAZR (a) mean Doppler velocity and (b) radar reflectivity factor for the 10 September 2017 event at ENA. (c) Cloud optical depth $\tau$ retrievals from the photometer (SPHOT, red) and radiometer (MFRSR, blue), with shaded (grey) regions indicating cloud samples used in comparisons for this study. Instantaneous retrieval uncertainty ranges for the SPHOT follows Chiu et al. (2012), whereas MFRSR uncertainty follows Min and Harrison (1996).**

Our Sc cloud criteria loosely follow previously published classification studies and guidelines (e.g., Remillard et al., 2012). To ensure appropriate samples from widespread Sc conditions, we required each event to contain a minimum of one hour of overlapping observations from the MFRSR and photometer. The MFRSR and photometer retrievals were only available during daytime hours, which also limited the times where these overlapping overcast cloud conditions were sampled. To designate "warm/low" Sc conditions, we applied additional filters based on collocated ARM profiling products. First, the multi-sensor Active Remote Sensing of Clouds (ARSCL) product was used to estimate cloud boundaries and other radar quantities at 4 s time and 30 m height intervals using a multi-sensor approach (e.g., Clothiaux et al., 2000). The warm/low boundary layer criteria were partially enforced by only considering clouds with mean cloud echo top heights (CTH) as estimated by ARSCL below 4 km (with no higher cloud layers also observed above that altitude). Second, ARM's linearly interpolated sounding product was also used to remove events where these same ARSCL cloud layers extended above the melting level. The resulting set of events can most often be characterized as single-layer Sc cloud decks, with the findings not significantly altered when multi-layer clouds were present.

This study did not consider samples that were associated with measurable surface precipitation. To help remove rainy conditions or problematic comparisons therein, the mean cloud base designated by ARSCL must exceed 0.5 km AGL during the intervals, while surface rain gauge and downward Ka-band ARM Zenith Radar (KAZR, e.g., Kollias et al., 2020) mean Doppler velocity in the lowest 300 m exceeding 4 ms$^{-1}$ were also used to identify rain at/near the surface. Still, our criteria





allow for Sc with in-cloud drizzle and subcloud virga (i.e., sampling the presence of radar reflectivity echoes below ARSCL cloud base). For virga conditions (where precipitation was assumed to evaporate completely before reaching the ground), we kept such retrievals in our evaluations since we assumed any raindrops were only present in small number concentrations and had limited impacts on the radiometric quantities used in the retrieval method. However, we excluded observations under conditions with precipitation measurable by surface rain gauges, since the dome of the radiometer would become wet, implying the measurements may be contaminated or less reliable. We also ignored any samples for which any of the instruments retrieved LWP > 400 g m$^{-2}$, as our own visual inspection suggested such larger values typically occurred near rainy conditions.

In total, our dataset contains 36 qualifying events over the SGP site, and 80 qualifying events over the ENA site. There were 855 (SGP) and 1341 (ENA) 5-minute observations that met our Sc criteria. Qualifying events at the SGP site were typically associated with post-frontal stratocumulus conditions (e.g., Kollias and Albrecht, 2000; Mechem et al., 2010). These events were collected during the spring and fall seasons where frontal intrusions at SGP are most common (approx. 90% of our dataset). A wider seasonal cloud distribution was collected at ENA, with the summertime months (June through August) being the most common for observations (approx. 35%). The fewest Sc events were collected between December and February (approx. 15%). This seasonal bias was expected given our focus on warm Sc conditions having CTHs below the melting level. For ENA, the dataset mean CTH was 1583m ± 375m (suitably below the imposed 4 km top), while SGP post-frontal CTHs were slightly higher and more variable, 1827 m ± 782 m.

## 2.2 Cimel Sunphotometer and its Automated Cloud Property Retrievals

The Cimel sunphotometer is a ground-based scanning photometer for passive remote sensing of the atmosphere, with NASA AERONET calibrating and maintaining these instruments, while processing certain data as part of their global archive. The ARM user facility deploys its photometers at three fixed sites and offers mobile deployments on request. The photometer "cloud mode" has been employed by ARM since 2007. During this mode, the instrument points to zenith and obtains high gain sky mode observations of radiance in at least six of its 9 channels: 380 (newer CE318T models), 440, 500, 675, 870, 1020, and 1640 nm wavelengths. Although the instrument requires less than 5 minutes to cycle through these channels, the availability for scheduling "cloud mode" retrievals is limited by the overall photometer sequencing and contingent on the solar zenith angle and instrument model. For much of this data record, retrievals were performed at 15-minute intervals (prior to October 2017 at SGP, February 2021 at ENA) when environmental conditions allowed, while new models improved availability to 5-minute updates when not operating in any of the other observing modes.

The automated retrievals we implement use zenith radiance measurements at 440, 870, and 1640 nm wavelengths. This approach simultaneously retrieves $\tau$ and $r_e$, with these quantities used to compute LWP as:



$$\text{LWP} = \frac{2}{3}\, \rho_w\, \tau\, r_e \qquad [\text{g m}^{-2}], \quad (1)$$

where $\rho_w$ is the density of water, $r_e$ is in [μm], $\tau$ is unitless, and the expression in (1) assumes that liquid water content is
constant in the vertical (Stephens, 1978). The inputs to the algorithm are the calibrated photometer zenith radiance measurements and surface albedo estimated from the Terra and Aqua Moderate Resolution Imaging Spectroradiometer (MODIS, "MCD43A2 and "MCD43A3" products, e.g., Schaaf et al., 2002). While this algorithm has been documented by Chiu et al. (2010; 2012), aspects for its implementation are briefly summarized below.

The ground-based zenith radiance for clouds at a given wavelength may be expressed as functions of the incoming radiance, the cloud $r_e$ and $\tau$, and the albedo of the underlying surface. By including the 1640 nm water-absorbing wavelength, Chiu et al. (2012) three-channel constraint methods enabled $r_e$ estimates since the zenith radiance behavior for 1640 nm decreases with droplet size due to absorption, whereas radiances at 870-nm increase due to forward scattering. In practice, retrieval sensitivity of zenith radiance measurements to larger droplet size, as well as other practical limitations for radiance and surface albedo
estimates, may conspire to undermine the usefulness of this third channel for $r_e$ retrievals. To combat the diminishing nature of those effects, Chiu et al. (2012) implemented a multi-step perturbation approach to assess retrieval uncertainty. This approach first considers a 5-10% uncertainty (normally distributed, input sensitivity) in zenith radiance and surface albedo measurements. The perturbed zenith radiances are subsequently compared to a calculated look-up table computed from the discrete-ordinate-method radiative transfer model (DISORT; Stamnes et al., 1988) over input ranges typical for ARM sites
(e.g., Zhao et al., 2012, 2013).

This implementation follows Chiu et al. (2012) by defining a solution from the photometer retrieval as "viable" when the zenith radiances agree with the look-up table to within 10% at the 440, 870 nm wavelengths. Any viable solutions are sorted based on errors in the zenith radiance at the 1640 nm, with the five best solutions (i.e., smallest errors) averaged to generate a
single solution for the set of the perturbed zenith radiance and surface albedos. Chiu et al. (2012) recommended this procedure be repeated 40 times using randomly generated perturbations. Reported retrievals for $\tau$ and $r_e$ are obtained by taking the mean of 40 repetitions. Sensitivity tests (not shown) that considered additional perturbations did not produce significant changes in the retrieved quantities.

This perturbation uncertainty (defined here as calculating the standard error) is reported by these photometer retrievals (herein, SPHOT) as its instantaneous retrieval uncertainty. For our dataset, the average values for these reported uncertainties at ENA in $\tau$ and $r_e$ estimates are 1.19 (unitless) and 2.1 μm, respectively. For the SGP dataset, these uncertainties are 1.56 (unitless) for $\tau$ and 1.46 μm for $r_e$. These values may also be reported as relative errors at a level of 5-10% of the reported $\tau$ estimates, or 15-20% of the reported $r_e$ estimates.



## 2.3 Additional ARM Cloud Property Retrieval VAPs


Comparisons are performed against related ARM products common to both sites. The primary comparison is with the MFRSR products that apply an iterative approach to compute $\tau$ and estimates $r_e$ using an independent LWP estimate. The reported instantaneous retrieval errors for these MFRSR products are on the order of 5% and 25% for $\tau$ and $r_e$, respectively (e.g., Min and Harrison, 1996; Min et al., 2003). These products require an LWP estimate, and this LWP estimate is taken from the ARM

Microwave Radiometers (MWR, 5.9° FOV). Herein, we refer to MWR retrievals for LWP according to the ARM naming "MWRRET", which is a product available at a 20 s time resolution. If an LWP estimate is unavailable from MWRRET, the MFRSR retrievals assume a fixed $r_e$ value ($r_e$ = 10 μm), and the associated product returns only a value for $\tau$ (Turner et al., 2004). This study avoids null instances and only compares retrieved quantities in cases where non-zero LWP estimates are available. LWP estimates obtained from the MWR in previous marine studies are typically reported with uncertainties $O$[10-

20 g m$^{-2}$] (e.g., Cadeddu et al., 2023). These MWRRET retrievals assume all its liquid media is in the Rayleigh scattering regime. It has been found this approach overestimates LWP retrievals (> 10%) at ENA in the presence of larger drizzle hydrometeors when LWP exceeds 100-200 g m$^{-2}$ (e.g., Cadeddu et al., 2023). For a secondary LWP reference, we consider LWP estimates from the Tropospheric Optimal Estimate approach (TROPoe, Turner and Löhnert 2014, 2021; Turner and Blumberg, 2019) that were selectively available at both sites starting in 2016. These methods use atmospheric emitted radiance

interferometer observations and assume single-layer clouds.

The "baseline ARM retrieval of cloud microphysical properties" product (herein, "MICROBASE", e.g., Dunn et al., 2011; Zhao et al., 2012; Huang et al., 2012) was also available to evaluate relative $r_e$ retrieval performance. The MICROBASE algorithm uses cloud radar reflectivity Z estimates from the KAZR, along with LWP estimates from its MWRs and temperature

profiles from soundings. From these inputs, MICROBASE estimates the profiles for the liquid water content (LWC) and liquid $r_e$ if the estimated LWP is positive/non-zero. The retrieved LWC follows Liao and Sassen (1994), as:

$$\text{LWC} = \left[\frac{N_0 Z}{3.6}\right]^{5/9} \qquad [\text{g m}^{-3}], \qquad (2)$$

where $N_0$ is a constant reference cloud number concentration = 100 cm$^{-3}$, $Z$ is the equivalent radar reflectivity factor $Z$ (for KAZR, at 35 GHz) in linear [mm$^6$m$^{-3}$] units. The LWC is scaled by the ratio of the estimated LWP (from the integration of the LWC in the column) and the LWP retrieved from the MWR. Owing to this scaling, relative radar $Z$ calibration offsets or corrections do not strongly influence these retrievals.

The MICROBASE effective radius estimates follow Frisch et al. (1995):



$$r_e = \frac{e^{2.5\sigma^2}}{\left[\frac{4}{3}\pi\rho_w N \, \text{LWC} \, e^{9\sigma^2/2}\right]^{1/3}} \qquad [\mu m], \tag{3}$$

where $N$ is a constant cloud particle number concentration CDNC = 200 cm$^{-3}$, $\rho_w$ is the density of water, and $\sigma$ is the width of

a log-normal droplet distribution = 0.35. The MICROBASE retrievals provide an uncertainty estimate for each of its cloud microphysical retrievals based on a perturbation analysis performed for typical ranges of its input parameters (Zhao et al., 2014). The relative errors that MICROBASE products report in applying Eq. (2) and (3) are estimated by perturbing their inputs (i.e., LWP). Zhao et al. (2014) reported this uncertainty in $r_e$ at 5%.

While these reported $r_e$ errors are lower than the claims from our collocated SPHOT retrievals, these estimates did not consider physical process uncertainty that stems from violations to above MICROBASE assumptions. For example, those standard products estimate $r_e$ by assuming a default CDNC = 200 cm$^{-3}$, a representative value for all-sky conditions at the SGP site. This value may be appropriate for midlatitude continental Sc, and its selection was informed as part of radiative closure studies performed at SGP. This CDNC value may be significantly larger than expectations for marine Sc conditions $O$[50 cm$^{-3}$] (e.g.,

Wood et al. 2015; Bennartz and Rausch, 2017; Wang et al., 2022). To test this assumption at ENA, we modified this retrieval to include an additional fixed CDNC = 50 cm$^{-3}$ assumption. The authors also implemented a lower CDNC = 100 cm$^{-3}$ test for SGP, as this CDNC is consistent with prior studies that contributed to MICROBASE development (e.g., Liao and Sassen, 1994). One physical argument for the need to consider a lower CDNC at SGP is that Sc cloud conditions that predominantly form within the hours following frontal passages may be associated with reduced aerosol.


Finally, there is ambiguity when defining an optimal strategy to compare our bulk, surface-based SPHOT retrievals to time-height MICROBASE $r_e$ profiles. We consider the average 5-minute in-cloud time-height $r_e$ estimates from the MICROBASE profiles. We also record the maximum MICROBASE value from within those clouds as a second reference. This is done for simplicity, as MICROBASE retrievals assume single-layer clouds, but allow for inhomogeneous cloud $r_e$ profiles that yield

different averaged behaviors in lower or upper parts of these clouds.

## 3 Results and Discussion: SGP Stratocumulus Clouds

This section summarizes results for SPHOT retrievals of cloud properties as collected from the Sc events over the SGP site. Data were drawn from 36 qualifying events that yielded 855 5-minute comparisons between the various retrievals. For these Sc datasets, 26 events were associated with 645 additional comparisons from collocated TROPoe retrievals for LWP.





### 3.1 SGP Cloud Optical Depth, Effective Radius Retrievals and Liquid Water Path Estimates

In Fig. 2, we plot scatterplots (Figure 2a) and box-whisker displays (Figure 2b) for the SPHOT and MFRSR $\tau$ retrievals. Overall, $\tau$ comparisons between these instruments were associated with the highest coefficient of determination ($R^2 = 0.7$). As also reported in Chiu et al. (2012), retrievals from the SPHOT are larger than those from MFRSR measurements. For perfectly homogeneous clouds, retrievals from these two methods should be identical. However, clouds are never homogeneous and SPHOT $\tau$ will typically be larger than what is reported by MFRSR retrievals. This is because the transmitted flux measurement by the MFRSR monotonically decreases with increasing $\tau$, following a convex curve (as shown in Figure 3). For inhomogeneous clouds, the MFRSR measures an averaged flux that will always correspond to an equal or smaller $\tau$ than the average obtained from the $\tau$ values inferred from a SPHOT. The magnitude of this systematic difference between MFRSR- and SPHOT-based retrieval depends on the degree of cloud inhomogeneity.

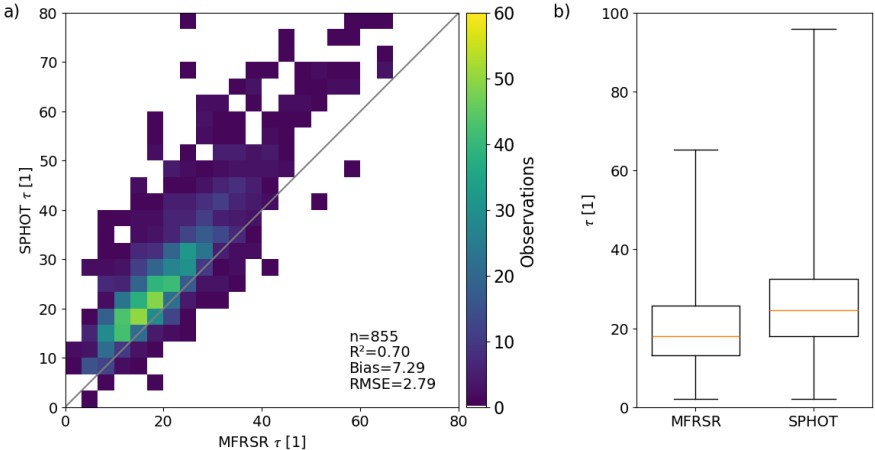

**Figure 2: (a) Scatterplot of cloud optical depth $\tau$ retrievals [unitless] from the SPHOT and MFRSR. (b) Associated SPHOT and MFRSR box and whisker plots for $\tau$ distribution quartiles and extremes, with distribution medians in yellow.**





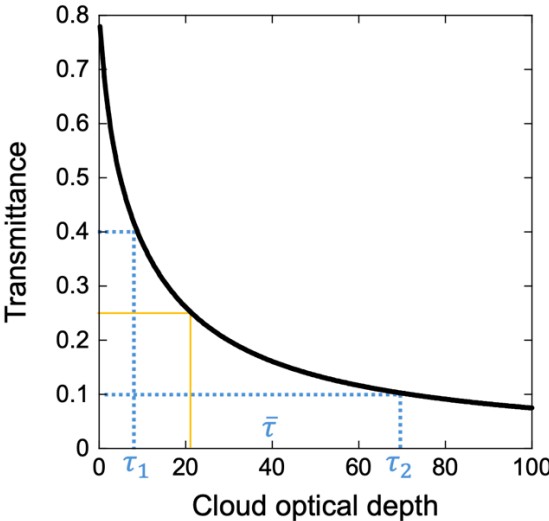


**Figure 3: Example relationship between the transmittance and cloud optical depth ($\tau$), as denoted by the solid black curve. Suppose an inhomogeneous scene containing two optical depths (e.g., $\tau_1$ and $\tau_2$ that lead to transmittances of 0.4 and 0.1, respectively), shown by dotted blue lines. Due to its hemispherical FOV, the MFRSR would measure an averaged transmittance (i.e., 0.25) and a retrieved $\tau$ of ~20 (orange lines). In contrast, SPHOT with a narrow FOV would retrieve $\tau_1$ and $\tau_2$, leading to an average $\bar{\tau}$ of ~40.**


For this study, we report Bias = $<\Delta>$ and the Root Mean Square Error RMSE = $(<\Delta^2>)^{1/2}$. The $\Delta$ is the difference between the SPHOT retrieval and the similar quantity from the reference instrument. Overall, $\tau$ retrieval distributions indicate a higher SPHOT median (24.5) to the MFRSR (18.1), and extended quartiles/tail. The RMSE = 2.79 (unitless, relative error > 10%) between these units is larger than retrieval uncertainty typically reported for either instrument with respect to algorithm 265 perturbation tests ($\tau$ error < 2, relative error < 5%).

SGP retrieval intercomparisons for $r_e$ are summarized in Fig. 4. In Fig. 4a, we plot relative performance when compared to the default MICROBASE implementation, while Fig. 4b plots the MICROBASE retrieval behavior if applying a modified assumption for fixed CDNC = 100 cm$^{-3}$. In Fig. 4c, we plot a reference maximum $r_e$ from the default MICROBASE during 270 each 5-minute sample. Summary box and whisker displays for all $r_e$ estimates are plotted in Fig. 4d.





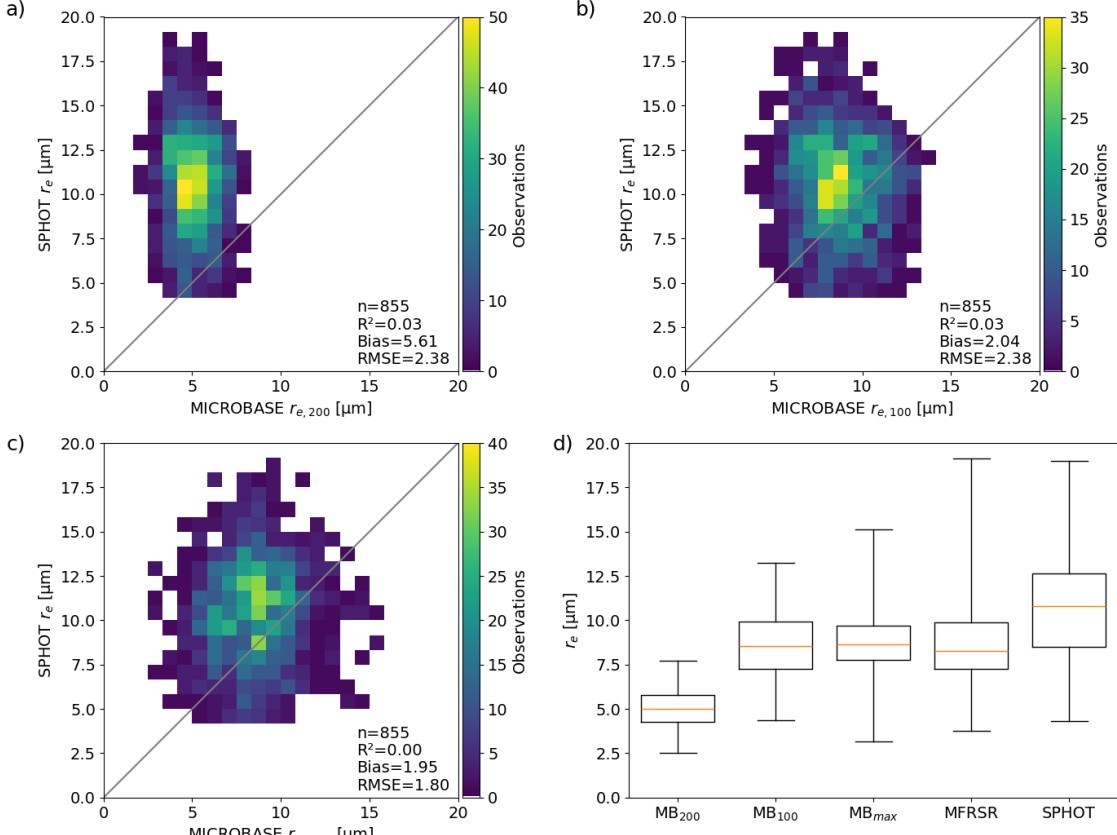

**Figure 4: Scatterplots of $r_e$ retrievals with associated R², Bias and RMSE[μm] from the SPHOT and MICROBASE with (a)**
**MICROBASE CDNC assumption of 200 cm⁻³, (b) MICROBASE CDNC assumption of 100 cm⁻³, (c) selection of the maximum**
**MICROBASE $r_e$ in the column (using CDNC = 200 cm⁻³). (d) SPHOT, MICROBASE, and MFRSR box and whisker plots for $r_e$**
**distribution quartiles and extremes, with distribution medians in yellow.**

Overall, relative SPHOT-MICROBASE comparisons highlight a shift towards lower bias when MICROBASE retrievals at

SGP adopt the lower fixed CDNC. However, most pairings exhibit negligible retrieval correlations, with this low-bias CDNC

= 100 cm⁻³ configuration reporting an R² < 0.1 and a RMSE of 2.38 μm. SPHOT retrieved $r_e$ median value(s) and quartile

distributions skew larger than ranges estimated by the other instruments (median $r_e$ = 10.81 μm), with the largest discrepancies

found between SPHOT and MICROBASE's default CDNC = 200 cm⁻³ (median $r_e$ = 5.01μm). Some offset between

MICROBASE and SPHOT should be expected, since both retrievals cannot properly attribute and distribute profile water

content in the presence of drizzle. In the case of MICROBASE's $r_e$ estimates, the LWC is attributed to a cloud distribution

(i.e., assumes a larger number of only slightly larger drops) rather than attributing this liquid to a few larger drizzle droplets at

the expense of small particles. Similarly, SPHOT $r_e$ will be skewed high in the presence of drizzle, as absorption at the 1640



nm channel will increasingly act towards solutions having larger $r_e$. Previous studies suggest a critical effective radius for drizzle onset may be associated with a cloud top $r_e$ that exceeds 12 μm (e.g., Rosenfeld et al., 2012). Considering such

statements as one guideline, median SPHOT properties suggest common presence of drizzle within SGP samples.

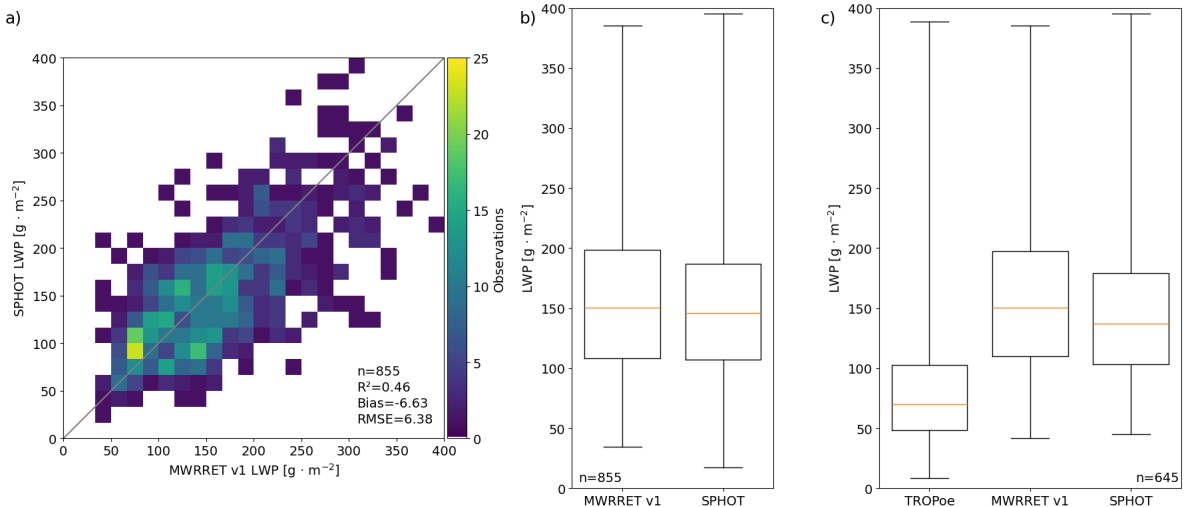

**Figure 5: (a) Scatterplot of LWP retrievals and associated R², Bias and RMSE [g m⁻²] from the SPHOT and MWRRET ("Version 1"). Box and whisker plots for LWP distribution quartiles and extremes, with distribution medians in yellow for (b) cumulative**
**SPHOT and MWRRET datasets, and (c) datasets including TROPoe product availability.**

In Fig. 5a, we plot a scatterplot for the LWP estimated by SPHOT following Eq. (1) versus estimates from the MWR. In Fig. 5b and 5c, the cumulative distributions for these comparisons are provided, as well as LWP comparisons when TROPoe products were available. There is solid agreement between the SPHOT and "Version 1" of the MWRRET LWP estimates, with

$R^2 = 0.46$ and RMSE of 6.38 g m⁻². At SGP, ARM offers two versions of MWR estimates, with a "Version 2" utilizing the MWR's 89 GHz frequency to better separate the contributions of drizzle to the LWP. Note that the MWRRET "Version 2" products (not shown) and TROPoe LWP estimate comparisons indicate a lower median and narrower spread of LWP estimates (we find a median value of 70.2 g m⁻² from TROPoe compared to 136.9 g m⁻² from SPHOT). An explanation for these discrepancies is the presence of drizzle and observed shifts that occur at higher-relative $r_e$ levels. This high bias is common at

SGP, even for clouds having lower $r_e$ less consistent with drizzle at sites such as ENA. We suggest this may be tied to the wider spread and elevated CDNC conditions allowing higher $\tau$ and/or LWP that are consistent with drizzle presence at lower $r_e$.

## 3.2 Discussion of Cumulative SGP Sc Retrieval Performance

While SPHOT retrievals at SGP are found in modest agreement with the collocated sensors – with an emphasis on non-
drizzling Sc conditions – it is important to consider potential controls on retrieval variability. In Fig. 6, scatterplots from the



previous section have been sorted according to a second retrieved quantity. In Fig. 6a and 6b, we plot $\tau$ performance indexed according to their associated LWP and $r_e$ retrievals, respectively. As before, SPHOT overestimates the values obtained from the MFRSR, with larger $\tau$ associated with larger LWP as expected. Most discrepancies we observe are physically consistent, i.e., higher offsets in $\tau$ are associated with samples where SPHOT retrieved smaller $r_e$ (i.e., $r_e$ values < 7-8 μm). Large offsets were observed for cases with moderate LWP $O$[150-200 g m$^{-2}$] (Figure 5a).

In Fig. 6c and 6d, we plot $r_e$ retrievals indexed according to $\tau$ and LWP. The largest discrepancies between SPHOT and the MICROBASE techniques for $r_e$ estimates are found at lower values of $\tau$ and within an intermediate range of LWP values where the SPHOT $r_e$ can be significantly larger. For larger optical thicknesses and LWP, the effective radius estimated by MICROBASE can be much larger than that from SPHOT. This behavior aligns with retrieval susceptibility to larger errors near conditions of drizzle onset. Overall, the SPHOT retrievals are higher than our adjusted MICROBASE estimates for $r_e$, consistent with previous findings (i.e., Figure 4d). This follows as the MICROBASE approach is unable to partition LWP to $r_e$ appropriately under drizzling conditions. Breakdowns for the estimated LWP from the SPHOT (Figure 6e-f) indicate that increasing LWP scales with increasing $\tau$, while LWP offsets are relatively unbiased or insensitive to the bulk magnitude of $\tau$ estimates. However, lower magnitude $r_e$ values from the SPHOT are those that occupy the relatively lower-biased LWP estimate space, and larger $r_e$ values are associated with relative SPHOT overestimates of LWP. This response is physically consistent with Eq. (1), with larger $r_e$ shifts (i.e., presence of drizzle) compensated by smaller, yet important shifts in $\tau$ estimates (high bias, yet commensurate with drizzle). This argument may also be consistent with SPHOT and MWRRET "Version 1" versus "Version 2" performances relative to the TROPoe approach with respect to the role small amounts of drizzle drive on LWP disconnects between these instruments.



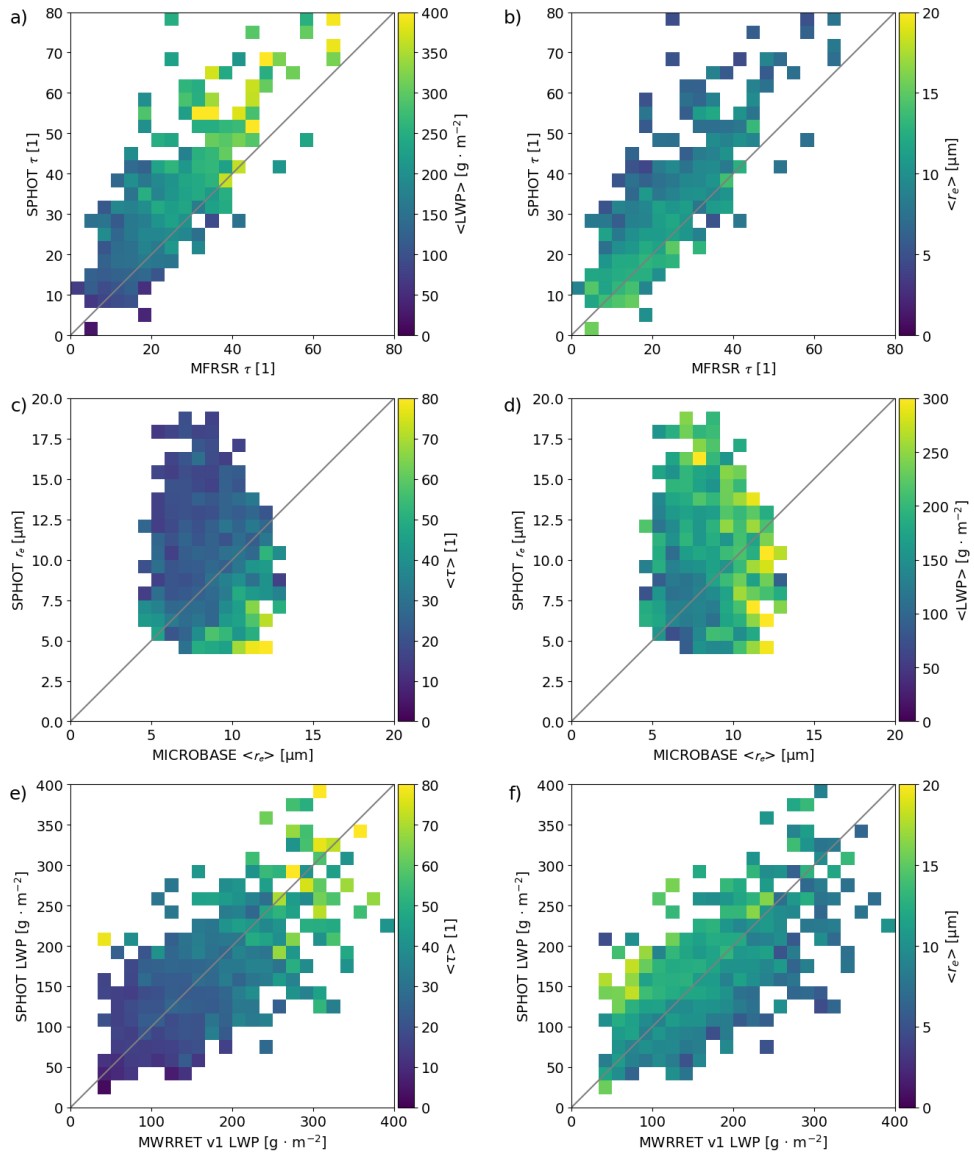

**Figure 6: For SGP events, (a, b) scatterplots of SPHOT and MFRSR $\tau$ retrievals contingent on $r_e$ and LWP retrievals from the SPHOT. (c, d) SGP scatterplots of SPHOT and MICROBASE $r_e$ retrievals contingent on $\tau$ and LWP retrievals from the SPHOT. (e,f) SGP scatterplots of SPHOT and MWRRET "Version 1" LWP retrievals contingent on $\tau$ and $r_e$ retrievals from the SPHOT.**

In Fig. 7, we plot the differences in LWP estimates between SPHOT and TROPoe products as a function of the $r_e$ estimated by the SPHOT. SPHOT LWP discrepancies compared to TROPoe estimates are exacerbated for $r_e > 10$ μm where there is a shift in SPHOT to relative overestimates of LWP. Drizzle onset is suggested as one factor in these discrepancies, but not the sole explanation for the LWP estimate offsets. Select overestimation of the LWP by the SPHOT method may also be attributed



to spatial inhomogeneity captured differently by the narrower FOV instrument that also promotes higher $\tau$ when compared to the MFRSR.

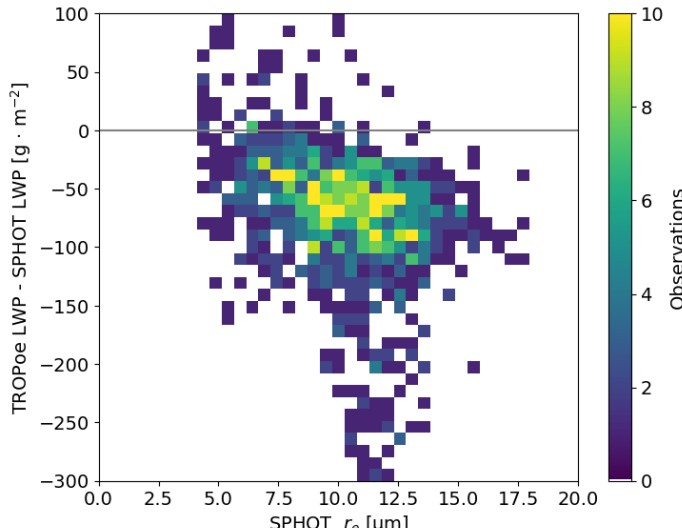

**Figure 7: Scatterplot for the difference (TROPoe retrieval minus SPHOT retrieval) in LWP estimates as a function of the associated**
**SPHOT $r_e$ retrieval estimate.**

In Table 1, we list SPHOT breakdowns for SGP clouds contingent on ARSCL-estimated cloud thickness. Thicker clouds show increasing values for $\tau$ and LWP. An absence of a similar trend in $r_e$ behaviors may be tied to SGP variability in CDNC and drizzle propensity under higher LWP often consistent with drizzle for marine locations (e.g., Rosenfeld et al., 2012; Zhu et al.,
2022). The standard deviations for retrievals vary according to the thickness and drizzle likelihood, with thicker clouds associated with a larger retrieval spread. Standard deviations for $\tau$ are larger than the values from algorithm perturbation tests, yet of similar magnitude to the RMSE observed between collocated MFRSR and SPHOT estimates. $r_e$ estimates sorted according to cloud thickness bins share relative standard deviation variability comparable to the uncertainty claims from retrieval perturbation and RMSE discrepancies as before. Overall, these standard deviations suggest that averaging within
these cloud thickness bins reduces the event-scale physical process variability back to the native measurement limitations.

Each cloud thickness bin for SGP exhibits bulk LWPs exceeding 100 g m$^{-2}$, with $\tau$ values typically exceeding 20. For marine clouds, such values would be consistent with copious drizzle (e.g., Zhu et al., 2022). We report radar mean Doppler velocity averages in SGP Sc clouds (the last columns of Table 1) to potentially identify shifts in mean downward air motions suggestive
of drizzle. However, all values we observe at SGP as a function of cloud thickness are downwards, and far exceed average motions found for similar clouds/thickness at ENA. This is likely because SGP observations are collected during post-frontal conditions, with vertical wind shear and larger-scale flows that contaminate radar velocity estimates more than would be



expected for quiescent ENA Sc. These factors render simple uses of mean Doppler velocity at SGP far less informative to drizzle onset than forthcoming ENA examples.


**Table 1: A summary of SGP Sc cloud properties and SPHOT retrievals contingent on cloud thickness. The MDV refers to the average in-cloud mean Doppler velocity from KAZR over the sampling window.**

| Cloud Thickness [m] | # obs | $<\tau>$ [1] | STDEV($\tau$) [1] | $r_e$ [μm] | STDEV($r_e$) [μm] | LWP [g m$^{-2}$] | STDEV (LWP) [g m$^{-2}$] | MDV [m s$^{-1}$] | STDEV(MDV) [m s$^{-1}$] |
|---|---|---|---|---|---|---|---|---|---|
| 200-400 | 86 | 18.92 | 8.42 | 11.01 | 3.53 | 107.99 | 39.58 | -0.11 | 0.38 |
| 400-600 | 224 | 24.9 | 9.59 | 10.52 | 2.59 | 139.51 | 49.7 | -0.18 | 0.35 |
| 600-800 | 180 | 27.72 | 14.27 | 10.48 | 2.61 | 150.25 | 57.77 | -0.22 | 0.34 |
| 800-1000 | 118 | 30.89 | 16.41 | 10.84 | 2.8 | 169.66 | 64.24 | -0.19 | 0.36 |
| 1000-1200 | 89 | 32.6 | 16.41 | 10.12 | 3.35 | 168.64 | 73.88 | -0.28 | 0.35 |
| > 1200 | 157 | 29.88 | 11.86 | 10.98 | 3.23 | 171.81 | 57.18 | -0.2 | 0.32 |

## 4 Results and Discussion: ENA Stratocumulus Clouds

This section presents summary results as collected during qualifying marine Sc events at the ENA site. These data were drawn from 80 qualifying events that provided 1341 5-minute comparisons between the various instruments. For these datasets, 41 events with 627 samples were available from collocated TROPoe LWP estimates.

### 4.1 ENA Retrieval Performance

In Fig. 8-10, we repeat SGP comparison plots for ENA. One notable change is the inclusion of a MICROBASE CDNC = 50

cm$^{-3}$ to accompany the baseline CDNC = 200 cm$^{-3}$ retrieval. Similar trends with SGP are observed, including a high offset for $\tau$ from the SPHOT compared to the MFRSR (Figure 8a,b). For the ENA site, only "Version 2" of the MWRRET products were available. The Sc $\tau$ at ENA are lower, reflected by a dataset median value = 16.0 that is 2/3rds of the value from our SGP dataset. Adjusting the default MICROBASE assumption to CDNC = 50 cm$^{-3}$ provides agreement with collocated $r_e$ retrievals (Figure 9a,b). We find a relatively similar median $r_e$ near 10 μm that is comparable to values estimated for SGP. In terms of




LWP comparisons (Figures 10a,b,c), the lower $\tau$ clouds associated with a similar $r_e$ implies ENA as having lower LWP values than typical SGP Sc. Overall, LWP estimates suggest lower standard errors than those from SGP Sc events (Figure 10b). We attribute improved comparisons to ENA's lesser propensity for drizzle, where SPHOT and TROPoe retrievals are reporting LWP at similar levels for most $r_e$ values retrieved (Figure 11). This improvement may also be coupled to the lower attendant CDNC conditions and reduced variability in CDNC also improving relative comparisons. We still observe SPHOT methods

tend to overestimate LWP surrounding likely conditions with drizzle onset where bulk $r_e$ > 12 μm (Figure 11).

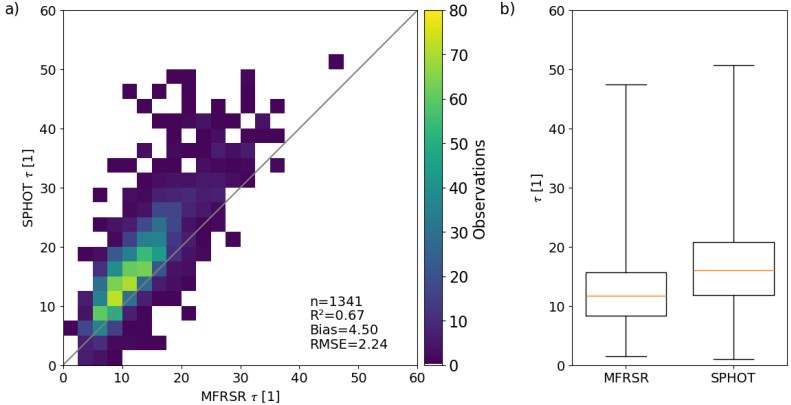

**Figure 8: As in Figure 2, but for ENA Sc $\tau$ samples from the SPHOT and MFRSR.**

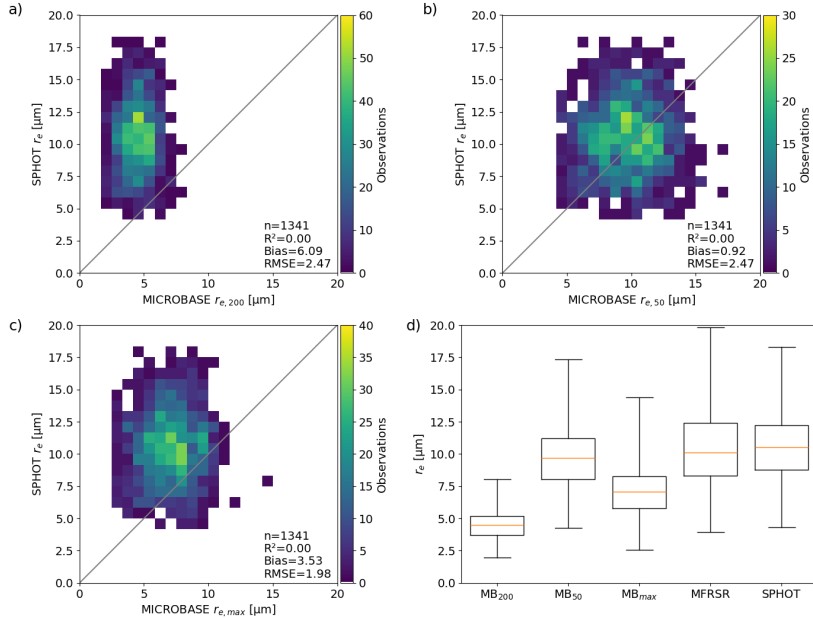


**Figure 9: As in Figure 4, but for ENA $r_e$ samples from the SPHOT, MICROBASE, and MFRSR. For the ENA dataset, a MICROBASE CDNC = 50 cm$^{-3}$ assumption has been substituted for the previous SGP assumption of CDNC = 100 cm$^{-3}$.**



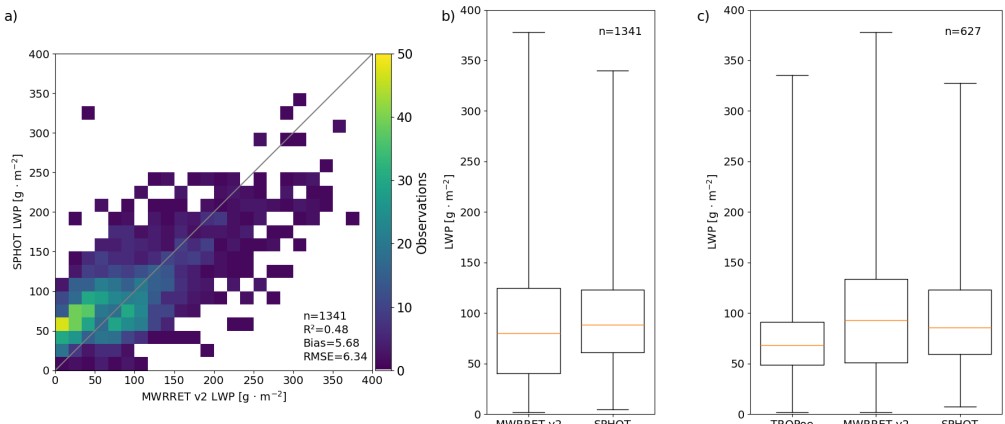

**Figure 10: As in Figure 5, but for ENA LWP estimates. For the ENA dataset, MWRRET "Version 2" retrievals have been substituted as "Version 1" is unavailable at the site.**

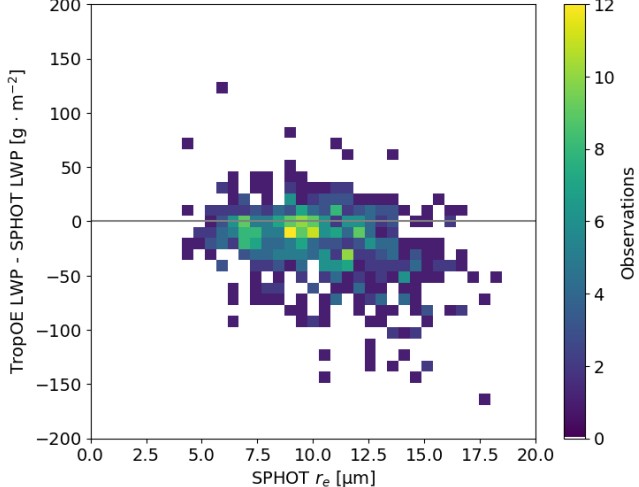

**Figure 11: As in Figure 7, but for ENA site.**


## 4.2 Discussion of Cumulative ENA Retrieval Performance

In Fig. 12, we plot ENA comparisons contingent on select jointly retrieved quantities. For these examples, we have included references to the KAZR in-cloud mean Doppler velocity averages during the 5-minute sampling window. For ENA, these estimates may serve as a better proxy for the presence of drizzle than what was found for post-frontal SGP Sc conditions. Summary retrieval performances as a function of cloud thickness for cumulative ENA clouds are found in Table 2. For these tables, we have included extra columns that report the number/percentage of ENA samples where ARSCL recorded reflectivity





factor Z below the ceilometer-estimated cloud base. These below cloud signatures may also act as a proxy for clouds with more substantial drizzle at ENA (e.g., Yang et al., 2018). Such signatures were not viable at SGP because insect contamination below the cloud often limits using these echoes without applying decluttering techniques (e.g., Williams et al., 2021).


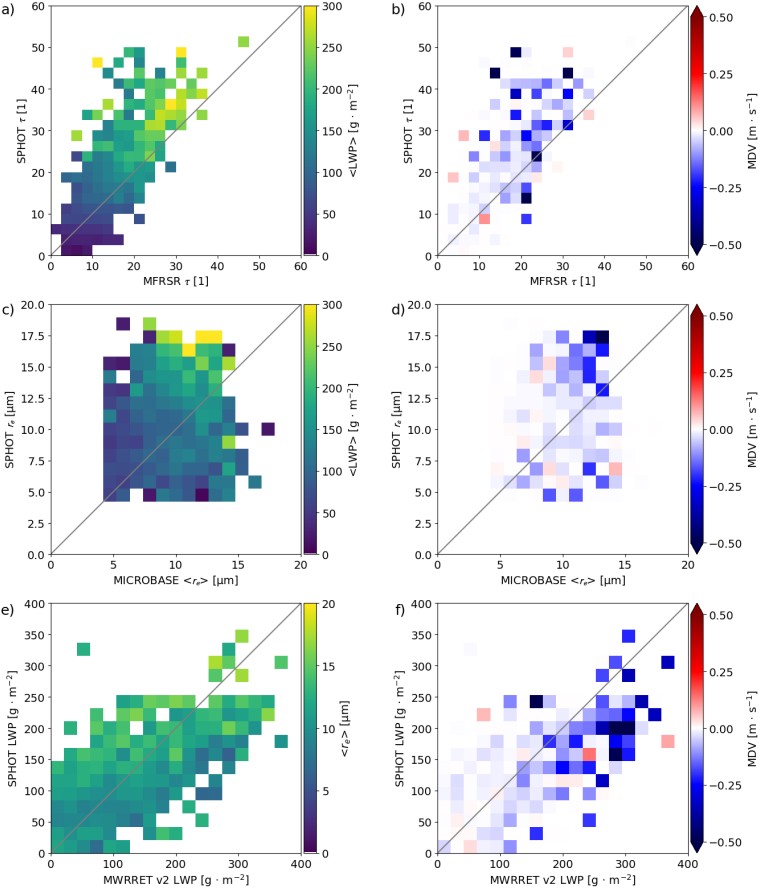

**Figure 12: For ENA events, (a, b) scatterplots of SPHOT and MFRSR $\tau$ retrievals contingent on LWP from the SPHOT and mean Doppler velocity estimates from KAZR. (c, d) ENA scatterplots of SPHOT and MICROBASE $r_e$ retrievals contingent on LWP from the SPHOT and mean Doppler velocity estimates from KAZR. (e,f) ENA scatterplots of SPHOT and MWRRET "Version 2" LWP**
**retrievals contingent on $r_e$ from the SPHOT and mean Doppler velocity estimates from KAZR.**

As observed with SGP events, higher $\tau$ estimates are associated with higher mean LWPs. The mean Doppler velocity signatures help demonstrate a pronounced shift within ENA clouds once sufficient drizzle is present (for cloud LWPs exceeding 100 g m$^{-2}$ and associated $\tau$ that exceeds 20). Larger $r_e$ values are also often associated with larger LWP, with clouds exceeding LWP of 200 g m$^{-2}$ typically exhibiting $r_e > 12$ μm and downward mean Doppler velocity consistent with drizzle (Figure 12b,d). These

properties are qualitatively consistent with SGP, where LWP estimates are higher under conditions where the SPHOT $r_e$ estimates are larger (Figure 12e). Overall, Doppler velocity characteristics suggest downward motions and drizzle within these ENA Sc clouds once LWP exceeds approximately 100 g m$^{-2}$ and/or bulk $r_e$ exceeds 11-12 μm (Figure 12f).





In Table 2, cumulative ENA SPHOT performances as a function of cloud thickness follow expectations from previous marine

Sc studies (e.g., Zhu et al., 2022). ENA clouds having a similar cloud thickness to SGP clouds indicate similar mean $r_e$ values, however ENA clouds indicate a reduced $\tau$ compared to SGP, and therefore reduced LWP following calculations in Eq (1). For example, a typical 500 meter ENA cloud thickness records a LWP of 80 g m$^{-2}$ with an $\tau$ of 15; the similar SGP Sc cloud is associated with large shifts in $\tau$ (25) and LWP (140 g m$^{-2}$). Physically, this follows from an expectation that SGP Sc are associated with double the CDNC value, though other differences between post-frontal and marine Sc may also contribute

(e.g., Mechem et al., 2010). For marine clouds, the prominent shift in the mean Doppler velocity and in the propensity for Z echoes below clouds is commonplace once relative LWPs exceed 100 g m$^{-2}$ or cloud thickness reaches 1 km. Approximately half of the Sc clouds with thickness greater than 1.0 km indicate drizzle below the cloud base (bulk cloud $r_e$ > 12 μm). All thickness bins at ENA report some percentage (approx. 5%) of cloud samples having subcloud drizzle signatures.

**Table 2: As in Table 1, a summary of ENA Sc cloud properties and SPHOT retrievals contingent on cloud thickness. BC columns refer to the number and % of samples having measurable KAZR reflectivity factor below the ceilometer-estimated cloud base from the ARM ARSCL VAP.**

| Cloud Thickness [m] | # obs | $<\tau>$ [1] | STDEV($\tau$) [1] | $r_e$ [μm] | STDEV($r_e$) [μm] | LWP [g m$^{-2}$] | STDEV (LWP) [g m$^{-2}$] | MDV [m s$^{-1}$] | STDEV (MDV) [m s$^{-1}$] | BC | BC % |
|---|---|---|---|---|---|---|---|---|---|---|---|
| 200-400 | 220 | 11.43 | 4.64 | 10 | 2.29 | 59.36 | 23.76 | -0.01 | 0.06 | 10 | 4.55 |
| 400-600 | 395 | 14.88 | 5.83 | 10.05 | 2.24 | 80.04 | 34.55 | -0.01 | 0.07 | 14 | 3.54 |
| 600-800 | 342 | 17.96 | 7.4 | 10.52 | 2.57 | 101.02 | 41.76 | -0.01 | 0.08 | 35 | 10.23 |
| 800-1000 | 201 | 21.23 | 8.36 | 10.96 | 2.65 | 125.83 | 50.47 | -0.03 | 0.09 | 49 | 24.38 |
| 1000-1200 | 74 | 20.69 | 7.25 | 11.97 | 3.01 | 132.58 | 47.64 | -0.04 | 0.16 | 38 | 51.35 |
| > 1200 | 103 | 26.59 | 9.44 | 12.04 | 3.43 | 171.47 | 64.45 | -0.18 | 0.29 | 75 | 72.82 |

## 4.3 Potential Island Influences on ENA Sc Properties?

Several recent studies discuss the potential role that islands, such as ENA's Graciosa Island and its terrain or waves emanating off the surrounding islands in the Azores archipelago, may have on influencing the clouds and precipitation over those islands




(Houze, 2012). These studies often do not place as much emphasis on the expectations for Sc and anticipated impacts on the likelihood of precipitation that may lessen the usefulness of profiling retrievals (such as ARM-style measurements collected on islands) as representative of wider open ocean Sc cloud properties. For an ENA site located on the northern coastline of the island, studies by Giangrande et al. (2019) have suggested the influence that larger-scale southerly flow may play on overall cloud thickness or drizzle/rain properties observed over an upwind site, whereas Ghate et al. (2021) also indicated shifts in sub-cloud turbulence from Doppler lidar observations on days when surface winds come from directions associated with island versus onshore ocean flows.

In Tables 3 and 4, we include an ENA breakdown for "island" versus "ocean" wind events, applying similar definitions to those found in Ghate et al. (2021). For this site, "ocean" winds are those from 315° to 90°. For cloud retrievals from this ENA dataset, the "island" cloud days exhibit slightly larger values of LWP and $\tau$ for relatively similar $r_e$; however, the differences we observe in mean $\tau$/LWP characteristics are within the standard errors for clouds of that given thickness, and within measurement uncertainty for most individual retrievals. There is a suggestion that clouds upwind under island flows may have a higher propensity for drizzle formation (if based on using LWP increase as a proxy). However, several corroborating signatures – such as the enhancements in mean Doppler velocity – may also be attributed to more frequent gravity waves at cloud level commonly observed around the islands with flow over terrain. Similarly, flow over the islands may be associated with higher aerosol and/or CDNC, offsetting higher LWP in terms of potential for drizzle onset.

Although our sampling that is contingent on cloud thickness is independent and non-sequential, using significance tests such the Student's t-test on any differences we observe is typically not appropriate for these applications given the skewness of the cloud property distributions we are sampling. Mann-Whitney significance tests (e.g., H. B. Mann. and D. R. Whitney, 1947) suggest that differences we observe in $r_e$, mean Doppler velocity, and/or LWP may be significant, but with the caution that these tests also provide occasional contradictory results (i.e., for select results in thickness bins). Overall, our current dataset suggests that the role of these islands on the retrieved Sc cloud properties (i.e., typically in assuming widespread clouds are already present at daytime) is inconclusive, potentially suggesting only minor enhancements in longer-track cloud averages that may be otherwise indistinguishable on an individual event/sample basis.





**Table 3: As in Table 2, a summary of ENA Sc cloud properties and SPHOT retrievals contingent on cloud thickness with surface winds from southerly island-influenced directions (from 90°E to 315° WNW).**

| Cloud Thickness [m] | # obs | $<\tau>$ [1] | STDEV($\tau$) [1] | $r_e$ [μm] | STDEV($r_e$) [μm] | LWP [g m$^{-2}$] | STDEV (LWP) [g m$^{-2}$] | MDV [m s$^{-1}$] | STDEV (MDV) [m s$^{-1}$] | BC | BC % |
|---|---|---|---|---|---|---|---|---|---|---|---|
| 200-400 | 117 | 12.54 | 4.5 | 9.57 | 2.16 | 62.95 | 23.11 | -0.02 | 0.08 | 9 | 7.69 |
| 400-600 | 173 | 15.64 | 6.21 | 10.58 | 2.24 | 88.53 | 35.53 | -0.02 | 0.09 | 12 | 6.94 |
| 600-800 | 141 | 18.32 | 8.33 | 10.52 | 2.42 | 102.68 | 44.19 | -0.02 | 0.12 | 22 | 15.6 |
| 800-1000 | 58 | 20.91 | 7.74 | 10.95 | 2.39 | 124.01 | 44.09 | -0.02 | 0.08 | 10 | 17.24 |
| 1000-1200 | 22 | 21.26 | 5.55 | 11.94 | 3.13 | 137.41 | 42.03 | -0.06 | 0.27 | 8 | 36.36 |
| >1200 | 21 | 26.04 | 9.73 | 11.61 | 3.43 | 160.83 | 55.96 | -0.15 | 0.26 | 13 | 61.9 |

**Table 4: As in Table 3, a summary of ENA Sc cloud properties and SPHOT retrievals contingent on cloud thickness. Surface winds for these samples are those from northerly oceanic-influenced directions (from 315°WNW to 90° E).**

| Cloud Thickness [m] | # obs | $<\tau>$ [1] | STDEV($\tau$) [1] | $r_e$ [μm] | STDEV($r_e$) [μm] | LWP [g m$^{-2}$] | STDEV (LWP) [g m$^{-2}$] | MDV [m s$^{-1}$] | STDEV (MDV) [m s$^{-1}$] | BC | BC % |
|---|---|---|---|---|---|---|---|---|---|---|---|
| 200-400 | 103 | 10.17 | 4.47 | 10.49 | 2.34 | 55.29 | 23.83 | 0 | 0.02 | 1 | 0.97 |
| 400-600 | 222 | 14.29 | 5.45 | 9.64 | 2.15 | 73.42 | 32.26 | -0.01 | 0.06 | 2 | 0.9 |
| 600-800 | 201 | 17.7 | 6.65 | 10.52 | 2.67 | 99.86 | 39.93 | 0 | 0.04 | 13 | 6.47 |
| 800-1000 | 143 | 21.37 | 8.6 | 10.97 | 2.75 | 126.57 | 52.82 | -0.03 | 0.1 | 39 | 27.27 |
| 1000-1200 | 52 | 20.45 | 7.84 | 11.99 | 2.97 | 130.54 | 49.68 | -0.04 | 0.07 | 30 | 57.69 |
| >1200 | 82 | 26.73 | 9.36 | 12.16 | 3.43 | 174.2 | 66.17 | -0.19 | 0.3 | 62 | 75.61 |






## 5 Conclusions

This study presents cloud properties and instrument comparisons performed following the automating of a set of retrievals from photometers. This effort documents longer-term breakdowns of key relative retrieval performances for $\tau$, $r_e$ and LWP within marine and continental Sc clouds. Overall, the performance at the fixed ENA and SGP sites provides confidence and
initial uncertainty references for 3-channel retrieval methods to be similarly automated for non-vegetated surfaces and remote locations where 2-channel retrievals were not previously viable. Key takeaways from this study are summarized as follows:

- Photometer $\tau$ retrievals are offset high when compared to retrievals obtained from the wider FOV MFRSR.
Photometer $r_e$ retrievals are offset high relative to MFRSR and MICROBASE estimates, primarily under conditions when drizzle is present. The LWP calculated from SPHOT $\tau$ and $r_e$ is relatively unbiased when compared to collocated LWP references under settings without drizzle and/or within lower or less complex CDNC contexts.

- Stratocumulus conditions at SGP and ENA exhibit substantially different $\tau$ and LWP magnitudes for similar bulk $r_e$ estimates, as attributable to differences in CDNC levels between these regions. The typical SGP Sc exhibits double
the $\tau$ and LWP as the one observed at ENA; For similar cloud thickness, the Sc clouds at ENA and SGP share comparable $r_e$ that highlight propensity for Sc at both locations to form drizzle.

- "Drizzle" signatures become increasingly apparent once bulk SPHOT cloud $r_e$ estimates exceed 11 $\mu$m. ARM's baseline MICROBASE retrievals were modified from their standard assumptions (CDNC = 200 cm$^{-3}$) to align with photometer $r_e$ retrievals. These changes suggest CDNC values near 50 and 100 cm$^{-3}$ for cleaner ENA and "post-
frontal" SGP conditions, respectively.

- Simple tests for "island" versus "ocean" wind conditions as a proxy for local ENA island controls on Sc properties were performed. While cloud conditions having flows over the "island" potentially promoted higher $\tau$, LWP or drizzle propensity for clouds of similar thickness (to significance testing standards), these enhancements were small and within typical instrument sampling/retrieval errors.


### Data and Code Availability

All ARM data including the "ARSCL", "MET", "SPHOTCOD2CHIU", "MFRSRCLDOD", and "MICROBASEKaPLus" named "value-added product" or "VAP" datasets used by this study can be downloaded at https://www.arm.gov/ (last access: 2 February 2025). These data and VAP code requests may be access through the ARM Data Center "Data Discovery" portal
found at: https://adc.arm.gov/discovery/#/.



## Author Contributions

JR, LM, DW, SG, CSH, and MW contributed to preparing ARM data products and troubleshooting therein. KS, SG, and JR planned the experiments and comparisons. LM, CC, and CSH collected, processed and helped troubleshoot sunphotometer instrument data and automatic retrievals therein. All authors contributed to the scientific discussion and to the writing of this paper.

## Competing Interests

The authors declare that they have no conflict of interest.

## Acknowledgements

This paper has been authored by employees of Brookhaven Science Associates, LLC, under contract DE-SC0012704 with the U.S. DOE. The publisher by accepting the paper for publication acknowledges that the United States Government retains a nonexclusive, paid-up, irrevocable, worldwide license to publish or reproduce the published form of this paper, or allow others to do so, for United States Government purposes. We acknowledge support from the Atmospheric Radiation Measurement (ARM) program, a user facility of the U.S. DOE, Office of Science, sponsored by the Office of Biological and Environmental Research. Additional support was from the Atmospheric Systems Research (ASR) program of that office. All ARM data sets used for this study can be downloaded at http://www.arm.gov and associated with several VAPs as previously noted in the "Data Availability" section. We also would like to thank the ARM's Data Center (https://www.arm.gov/data/) for their support, account access, and interactions therein. The authors would like to acknowledge the extended team efforts of ARM developers at Brookhaven National Laboratory working with science sponsors and mentors, including initial code conversions and design reviews performed by Tami (Toto) Fairless. Additional thanks from the authors is extended to Karen Johnson (BNL), Damao Zhang (PNNL) and Dave Turner (NOAA, GSL) for helpful discussions on this manuscript.

## Financial Support

This research has been supported by the US Department of Energy (grant no. DE-SC0012704 and DE-SC0021167). This project was supported in part by the U.S. Department of Energy, Office of Science, Office of Workforce Development for Teachers and Scientists (WDTS) under the Science Undergraduate Laboratory Internships Program (SULI).

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
