# Peer review of "Marine and Continental Stratocumulus Cloud Microphysical Properties Obtained from Routine ARM Cimel Sunphotometer Observations"

_EGUsphere, 2025_

## Author Response (AR1)

**Response to Reviewers:**

"Marine and Continental Stratocumulus Cloud Microphysical Properties Obtained from Routine ARM Cimel Sunphotometer Observations"

Sookdar et al. (2025) egusphere 2025-694

**General Comments from the Authors:**

We would like to thank the two Anonymous Reviewers and the Editor for their comments to encourage this manuscript and improve it to be suitable for publication in AMT. For our responses to Reviewer comments, we have combined "Anonymous Reviewer" replies into a single document. For the reply, we use blue italics font to differentiate author replies from Reviewer comments.

**Anonymous Referee #1**

This manuscript provides a comprehensive comparison of bulk stratocumulus cloud properties. Those properties include optical thickness —  $\tau$  and effective radius — r e retrieved using three different methods (sun photometer— SPHOT, the MFRSR radiometer, and the MICROBASE algorithm) as well as SPHOT-retrieved liquid water path — LWP, compared against microwave radiometer-based and multi-instrument retrieval (MWRRET and TROPoe, respectively), all of which are produced by the ARM user facility. With SPHOT being the primary focus of this study, the authors use 6 years of observational data from the ARM SGP and ENA sites and find that SPHOT generally tends to over-estimate τ, r e, and LWP relative to the other "reference" instruments, though the results are not as simple and uncertainties (which I suspect are somewhat underestimated) serve as a critical and interesting point of discussion. Assumptions concerning droplet number concentration embedded into the retrieval algorithms play a critical role in retrieval discrepancies, as demonstrated by smaller differences between some of the retrievals at the "ostensibly simpler" ENA cloud scenes or when assumed retrieval CDNC are modified. It is also interesting that the retrieval differences exceed the reported uncertainty (i.e., uncertainty propagation is insufficient at the very least), which also raises some potential structural uncertainties in some or all of the retrievals.

The manuscript is reasonably written, with some references missing. I think that the manuscript can be accepted for publication after minor revisions, although I do provide a few main (rather than major) comments below.

We thank this Reviewer for their helpful suggestions. We have incorporated several changes to the manuscript that we believe are consistent with the main Reviewer concerns. These changes include improving several parts of the text, adding relevant references and data set citations, and incorporating several corrections to figures at Reviewer suggestion.

**Main comments:**

- Before the manuscript conclusion, the authors discuss island effects confounding ENA observations, suggesting that while there are some indications of island-driven differences, they are typically within the retrieval uncertainty range (Sect. 4.3).
  - (1) I think that this is an interesting and valid argument, which should be highlighted in a dedicated manuscript, given the potential community interest (and pushback), and the fact that this result (with its important ramifications) is mentioned briefly in the abstract and is missing from the title (as I think it should since that is not the focus of the manuscript). I leave it to the authors to decide whether they wish to include this short section in the manuscript or reserve it for a future publication, likely in a different journal (ACP?) where the relevant reader pool for this topic is presumably much larger.
  - (2) If the authors decide to keep this section in the manuscript, the final sentence of the abstract needs to be toned down to have a language similar to the final bullet in the conclusions

    section.
  - (3) One of the reasons I discuss this topic the way I do, is that there at least several more articles exploring and analyzing these effects over ENA, so the authors should also refer to these studies in their discussion (e.g., Jeong et al. 2022; https://doi.org/10.1029/2022JD037021, Zheng et al., 2021; https://doi.org/10.5194/acp-22-335-2022), should they chose to keep this analysis in the manuscript.

We thank the Reviewer for the comments [1]-[3] and will address these combined. Our preference is to keep the section. We agree this topic could be explored separately; we have revised the text to better communicate that these topics are nontrivial. We are not claiming island influences are not significant (we can point the Reviewer to more extreme examples in the ENA record), but for the ENA Sc event conditions we considered, such influences typically fall outside photometer ability to recognize these influences on an event basis.

The authors did consider adding reference to Zhang et al. (2022). However, the reference (as far

as we can tell) suggests that a surface northerly wind would induce additional updrafts in ENA Sc (owing to a "cliff north of the ENA site"). The statement did not seem consistent with the ENA geography (see Figure). This is repeated, so it did not seem to be a typo. The authors may reach out for clarification, e.g., if those authors were implying the upslope near the coast.

- Many of the discussions throughout the manuscript rely on an r\_e threshold of 12 um as a proxy for drizzle and rain formation (virga since surface precipitation is mitigated to a large extent as part of the methodology here). This raises two issues with the results and discussion:
  - (1) In the methodology section, the authors state about virga that "we kept such retrievals in our evaluations since we assumed any raindrops were only present in small number concentrations and had limited impacts on the radiometric quantities used in the retrieval method". With the discussion of drizzle as the leading culprit for retrieval discrepancies, their methodology argument no longer holds. There are many, and potentially more impactful, reasons for retrieval differences (e.g., droplet dispersion, low radar sensitivity

to cloud droplets, the mentioning of which is lacking, etc.). The text should be revised such that the storyline, limitations, etc. are coherent.

We thank the Reviewer for this comment. We refer to "12 um", but agree its context could be better explained. We've improved our text in several places on how photometer observations compare with previous Re expectations. We suggest that photometer measurements (those useful to estimate Re) will not be as influenced by the properties of the upper levels of Sc clouds as the lower levels; there is a potential disconnect between how one interprets photometer retrievals to those from satellite or LES that focus on the near "cloud top" behavior. There is also a potential discrepancy between photometer-retrieved Re values and those from the previous literature that may be attributed (in part) to the scale of the observations and/or smearing of the cloud properties, i.e., differences in the FOV and averaging may not align with finer-scale LES expectations at cloud top. One location in the text where we communicate these ideas is:

"Previous studies suggest a "critical" effective radius for drizzle onset as associated with a cloud top  $r_e$  of "at least 12" to 14 um (e.g., Rosenfeld and Gutman, 1994; Lebsock et al., 2011; Rosenfeld et al., 2012). Considering those previous statements are associated with observations or modeling performed near cloud top and/or at finer resolution, we use such values only as a guideline that the median SPHOT properties we retrieve of 11 um suggests the common presence of drizzle within our SGP samples. As introduced above, our temporal averaging combined with the FOV of the observations may result in our sampling of lower  $r_e$  retrieval values that are consistent with drizzle in the cloud above. Moreover, since surface-based photometer measurements are less constrained to the upper-levels of Sc when drizzle is forming (i.e., the observed radiance at the surface is dictated by the entire cloud layer), previous cloud top statements specific to cloud top may be better suited to intrinsic drizzle onset in  $r_e$ , whereas photometer measurements should experience lag or partial influences as that drizzle falls through the cloud."

On inconsistency / removal of drizzle: The authors agree there was a poor use of the language, "had limited impacts on the radiometric quantities used in the retrieval method." Our study attempts to avoid water accumulating on the instrument(s); we allow for retrievals with drizzle present in the column/cloud above. We believe there is demand within our community for LWP properties in the presence of (some) drizzle. We feel it is important to report these behaviors as users often include periphery times. The Reviewer is correct that our original statement was not consistent. We have rephrased the section.

Similarly, ENA KAZR observations (i.e., mean Doppler velocity MDV) are sensitive to the presence of drizzle; one example for this is reflected in MDV fields showing an increase in downwards motions (radar estimates preferentially weighted / fall speeds by drizzle onset / larger media). KAZR quantities overall are quite capable in shallow "cloud" settings (typically, Z to -50 dBz at 1 km), however Z and MDV are quantities disproportionately influenced by drizzle once drizzle-sized

drops are present in the radar volume. As with photometer FOV arguments, using 5-minute averaged MDV properties from ARSCL (performed for the entire cloud over that window) implies averaged properties reflect different contributions from drizzle with cloudy regions than bulk photometer sampling (that is also less influenced by the cloud properties highest aloft).

(2) To my knowledge, Rosenfeld argues (in the referenced as well as other papers, e.g., Rosenfeld and Gutman, 1994) that 14 um is the critical r\_e (and not 12 um). While I agree that drizzle could be pretty common at SGP and ENA (not considering case filtering), the reliance on 12 um results in arguments no longer being valid if 14 um is used. For example, in I. 289, the following sentence should be toned down if the r\_e threshold argument is to be used, because now 14 um is quite above the 3rd SPHOT quartile. In I. 421-422, as another example, the reliance on the 12 um threshold is fragile. It is possible that those clouds are drizzling, but I'm not sure that this is necessarily the strongest argument. It could very well be, but my understanding of the figures is that things break down pretty gradually and not necessarily due to r\_e thresholds. It is not surprising, in that context, that when examining absolute errors such as in Fig. 7 and 11, we see increasing deviations.

We agree that our results as originally presented were not communicated optimally, and fall on the lower end of previously reported expectations. We have attempted to be more consistent with what our data shows and the use of those reference values in the revised text. For ENA, we may suggest the photometer "critical" Re is arguably closer to 13 um when considering the relative error plots and other radar observations more conservatively.

Rosenfeld's study points to a Lebsock et al. (2011) reference that we have included. Lebscok et al. suggest a wider (lower) range of Re, however done for a wider set of precipitating clouds. It is not clear (to the authors) that the "14 um" value should be the expected value for photometer retrievals; However, this value may be reasonable as an intrinsic "cloud top" value around drizzle onset. It is not clear LES bin-emulating simulation results adequately capture cloud processes (i.e., Endo et al., 2019) to accept these values as the standard reference. Nevertheless, we agree with the suggestion that our discussions should conservatively present the topics.

Lebsock, M. D., T. S. L'Ecuyer, and G. L. Stephens, 2011: Detecting the Ratio of Rain and Cloud Water in Low-Latitude Shallow Marine Clouds. *J. Appl. Meteor. Climatol.*, **50**, 419–432, <a href="https://doi.org/10.1175/2010JAMC2494.1">https://doi.org/10.1175/2010JAMC2494.1</a>.

Endo, S., Zhang, D., Vogelmann, A. M., Kollias, P., Lamer, K., Oue, M., et al. (2019). Reconciling differences between large-eddy simulations and Doppler lidar observations of continental shallow cumulus cloud-base vertical velocity. *Geophysical Research Letters*, 46, 11539–11547. https://doi.org/10.1029/2019GL084893

**Minor comments:**

I 13. efforts --> applied methods

Ok.

l 13 - remove 'collected'

Ok.

l 41 - capabilities --> methods

Ok.

I 45-46 - I disagree with this statement. The photometer does not provide LWP/tau/D, etc., but require some retrieval model to estimate those quantities. The same can be said about many instruments from radiometers to radars. One could argue that the photometer with its multispectral approach is more or less constrained, and provide arguments w/r/t operating wavelengths, FOV, etc.

Agree. We have revised this statement.

147 - Provide a reference for this photometer instrument (handbook, etc.)

We have added references to the associated files/DOIs.

Also added handbook reference to Gregory (2011).

Gregory, L. "Cimel Sunphotometer (CSPHOT) Handbook.", Jan. 2011. https://doi.org/10.2172/1020262

I 52 - "ARM's Sun-Sky-Lunar Multispectral Photometer" - I don't understand where ARM comes in this sentence.

Agree. Dropped.

178 - move this reference for ENA to 1, 67

Agree. Will include existing references of Wang et al. (2022) and Wood et al. (2015).

181-82 - minimal discussion/description of Fig. 1 is missing.

Ok. Added a revised description.

l 106 - provide a reference for the linearly-interpolated sounding product.

Added DOI reference to this VAP:

Fairless, T., Jensen, M., Zhou, Ainfeng, & Giangrande, Scott E (2021). Interpolated Sounding and Gridded Sounding Value-Added Products. https://doi.org/10.2172/1248938

I 106 - 4 km AMSL or AGL?

Added "AGL"; Performed relative to KAZR height (~30m). Defined in a response below.

I 109 - add "(not shown)"

Added.

112 - define AGL

Ok, as above.

I 130, 131 - height units can be removed before the ±

Removed.

I 149 - provide units for rho w. Is it possible that unit conversion factors are missing in eq. 1?

The reviewer is correct. The density was  $10^6$  g m-3, with r\_e input in meters instead of microns. Otherwise, using  $10^{-12}$  g/micron $^3$ , with r\_e kept in microns.

I 155-165 - I presume that some PSD assumptions (shape, dispersion, etc.) are baked in the Chiu et al. (2012) and DISORT LUTs in order to estimate r\_e values, is that correct? If so, elaborate, because, if exists, that is an essential component (and uncertainty source) in such retrievals. Such uncertainties could inflate the relatively small uncertainties reported below and explain site discrepancies, for example.

We have revised the text. The lookup tables were generated using a gamma cloud droplet size distribution with a shape parameter of 7, an assumption that we believe agrees well with observations (e.g., Pörtge et al., 2023). As demonstrated in Fielding et al. (2014,doi:10.1002/2014JD021742), the retrieval is generally not sensitive to cloud droplet number concentration. Instead, uncertainties from measurement errors, surface albedo, and cloud inhomogeneity tend to have a greater impact on the retrievals produced by this method.

Pörtge, V., Kölling, T., Weber, A., Volkmer, L., Emde, C., Zinner, T., Forster, L., and Mayer, B.: High-spatial-resolution retrieval of cloud droplet size distribution from polarized observations of the cloudbow, Atmos. Meas. Tech., 16, 645–667, https://doi.org/10.5194/amt-16-645-2023, 2023.

I. 160 – "may conspire to undermine" - recommend rewording

Modified.

I. 160 - combat --> mitigate

Ok.

I. 185 - MWR - (1) provide reference (2) to my knowledge, the 3-ch MWR covered nearly the entire study period at both sites (as suggested by the version 2 retrieval, which utilizes the 3 channels), and that instrument has a much smaller FOV <= 3.5 degrees.

Agree. Fixed FOV statements. Added references:

Morris et al. Microwave Radiometer (MWR) Handbook. 2019. 10.2172/1020715

Cadeddu, MP. "Microwave Radiometer – 3-Channel (MWR3C) Instrument Handbook." , Mar. 2021. https://doi.org/10.2172/1039668

I. 186 - provide a reference to MWRRET

Added reference to Turner et al. (2007).

Turner, D. D., S. A. Clough, J. C. Liljegren, E. E. Clouthiaux, K. Cady-Pereira, and K. L. Gaustad (2007), Retrieving liquid water path and precipitable water vapor from the Atmospheric Radiation Measurement (ARM) microwave radiometers, IEEE Trans. Geosci. Remote Sens., 45(11), 3680–3689.

I. 192-193 – Worth mentioning that above, 60 g m-2 or so, the MWR governs the TROPoe LWP values since the IR signal fully attenuated (provide reference). From our understanding of how TROPoe is implemented (personal communication with D. Turner), TROPoe does not use the 89 GHz channel; this is because of the concerns with calibration errors influencing retrieval quality. The ARM TROPoe code instead only uses the 23.8 and 30/31 GHz channels. This choice has an advantage for making retrievals more consistently applied across different ARM locations (older-generation MWRs). The authors do not claim to be experts or responsible for TROPoe and MWRRET retrievals; As the Reviewer has mentioned, there may be other factors that contribute to differences in MWR and TROPoe retrievals. This includes the different FOVs for the AERI and MWR, similar to FOV arguments above. Nevertheless, we agree with the Reviewer that IR dominates the solution for LWP less than 60-80 g m-2, e.g., IR is saturating around 60 g m-2 (e.g., Turner 2007, JGR, reference now added).

Turner, D. D. (2007), Improved ground-based liquid water path retrievals using a combined infrared and microwave approach, J. Geophys. Res., 112, D15204, doi:10.1029/2007JD008530.

I. 199 - remove "its"

Agree.

I. 214 - CDNC = 200 cm-3 - is it correct that MICROBASE has an internal inconsistency since for LWP CDNC = 100 cm-3?

This is a good question and one we did not intentionally overlook. Both reviewers have questions on "default" MICROBASE and how to interpret this as a reference.

The default MICROBASE product setting has not been modified by ARM since its initial production. Its key assumptions are not optimized for any one condition, but a "baseline" for a variety of SGP cloud conditions. This default setting has been reported as tuned using radiative closure studies. The details for these closure efforts do not appear to be well-documented in the literature discussing MICROBASE. Our assumption has been that tuning was done through the lens of the CDNC parameter, set to 200 cm-3. The product was never modified or "matched" to any other site, and ARM releases this product at other sites using the default SGP configuration.

For the LWC-Z mapping expression as from Liao and Sassen (1994): We believe its use is consistent within MICROBASE's approach on how to approximately partition LWP into discrete LWC intervals (height) by using radar reflectivity Z as a reference for where in this cloud one expects relative higher/lower LWC. This use is an assumed fit centered on a particular No (100 cm-3) condition. However, how this is applied in MICROBASE is arguably less important to its impact.

First, LWC  $\sim$  N\*D^3 and Z  $\sim$  N\*D^6, thus any change may introduce some discrepancies (i.e., shifts in how LWC might get placed) for the non-precipitating (smaller drop) times where the N control to LWC versus Z is more influential than D. But, the basic concept is that wherever there is a relatively large Z, it will be assigned a relatively large LWC. Yet, the total of all LWC is constrained by the LWP. Thus, this relationship does not create LWC in the column, it simply moves it around the cloud. The changes to Re are subtle (slightly higher in one place, slightly lower in another), and less significant to how Re retrievals (overall) are scaled by the primary CDNC control.

As Liao and Sassen (1994) note, [this expression is] "for estimating liquid water content only if the cloud droplet concentration No is known, [however] comparison with empirical relationships

suggests that a value of No  $^{\sim}$  100 cm-3 produces satisfactory results in a variety of liquid phase clouds." In Liao and Sassen, the expression was intended for "direct" estimates, whereas for MICROBASE, what is important is how LWC profiles are partitioned in height. MICROBASE Re values are averaged in height and time, further smoothing the role of that LWC  $\rightarrow$  Re mapping.

Overall, there was not much value or control in modifying a "new" MICROBASE relationship specific to multiple No conditions, given the path constraint for LWC and the averaging we perform. The authors included a line in the revised manuscript to note this discrepancy.

1. 220-221 - recommend sentence rewording - something is not clear here.

**Reworded.**

I. 242 - somewhat nitpicky, but an uppercase R is typically used for multi-variate comparison, whereas a lowercase is used for single variable comparisons, such as in this case.

**Agree. Fixed.**

I. 246-249 - That is a good discussion, but it should be noted that we assume no wind here, since high winds over the averaging period could influence the SPHOT in a similar manner.

Added reference to the associated Re (8 um) and channel (440) for this figure.

I. 299-303 - If Version 2 for MWRRET is available, which I understand it is, I think that the authors should present and discuss that, a better-constrained retrieval, unless there is a strong argument against it. Given the information in the text regarding the lower LWP in v2 compared to v1 (also consistent with the literature), Fig. 5c appears somewhat misleading at present. Agree. We did not use MWRRETv2 in SGP comparisons; This was an oversight, and the behaviors compared favorably to TROPoe. We now include MWRRETv2 examples in revised figures.

1 304-307 - confusing sentences. Consider rewording to deliver the bottom line more clearly.

**Re-worded the lines.**

I. 338-339 - I agree that we see many more outliers above 10 um, but I'm not convinced that, on average, the slope of the LWP diff increases above that threshold - a curve of diff vs. r\_e would help in Fig. 7.

Revised. We agree that Figures 7 and 11 may be plotted better as relative errors (suggested below), revised the plots and discussion according to Reviewer suggestions. However, the authors did not include a curve fit to the revised plots. We have plotted some examples for Reviewer

benefit below; we felt that adding curve(s) imposed a physical understanding to this that was less clear for the authors given the complexities of SGP Sc clouds (as compared to ENA, where a curve is seemingly not required). We felt it was simpler (conservative) to communicate that the SGP results are increasingly offset for larger Re (with our median samples having Re > 11 um), while ENA offsets do not overall suggest prominent relative errors until Re > 12 um, as associated with drizzle onset as a possible factor.

I 340-342 - could droplet dispersion be a factor here? I believe that we do not necessarily need to pass "the drizzling" threshold to argue it.

The SGP behaviors are difficult to interpret. Simpler ENA examples as in Fig. 11 and associated ENA radar obs provide confidence to attribute or point this shift (relative to photometer Re retrievals) to the presence of drizzle. We suggest the drizzle-influence shift to be associated with a photometer Re value of "at least" 12 um (to borrow Rosenfeld usage), though bulk values for SGP and ENA may be skewed lower for other reasons mentioned above. We are more confident in our radar interpretations and Re connections at ENA that skew towards higher-relative Re values nearer to 12-13 um. This ability to cross-confirm with MDV was not possible for SGP, owing to the more turbulent nature of those post-frontal Sc.

1354-355 - confusing sentence. Recommend rewording.

**Revised.**

363-364 - nice conclusion regarding context as essential for MDV-based drizzle determination, but keep in mind that the dataset examined here is drizzle-mitigated to some extent in the first place.

For SGP, insect contamination below the cloud limits this approach. Unfortunately, Doppler spectra methods such as the referenced Williams et al. (2021) do not "declutter" insect echoes, simply designate insects. "In-cloud" behaviors suggest that SGP preferentially showed

downwards air/media motions that were not consistent with expectations and/or ENA. This was also a bit surprising because a limited number of previous studies (Mechem, Kollias, etc.) seemed to suggest SGP Sc clouds (radar properties) as less turbulent. We do not think any SGP preferential downwards air motion we observed was related to KAZR radar not vertically pointing, though this could introduce biases under similar wind/Sc conditions otherwise.

1377 - 2/3rds -->two-third

**Fixed.**

I 406-407 - how far below the ceilometer cloud base would you consider it sub-cloud? There have been a few studies in recent years suggesting that ceilometers tend to detect cloud base several tens of meters above the "true" cloud base, so an offset of 100 m or so are required to get meaningful results. Remember that Ka-band radars are less sensitive to cloud droplets, especially smaller ones. Therefore, it wouldn't be surprising if the Ka-band radar doesn't detects any echoes below the reported ceilometer cloud base.

BTW, this lack of radar sensitivity to cloud droplets could be, at least in part, the source for the lack of MICROBASE sensitivity, which I think the authors should emphasize more. The MICROBASE retrieval appears quite concerning in general, and I'm surprised that there is no mention of its very weak instantaneous prediction skill (demonstrated by the various joint histograms, e.g., Figs. 9 and 12) anywhere in the conclusions. On that note, a MICROBASE-TROPoe comparison would be interesting.

Good comment, and encouraged the authors to reevaluate our methods for tracking these % and spot a deficiency in our coding and reporting of these %. Originally, the values were being estimated for when we were observing any echoes down close to the surface (approx. 340 m AGL), which implied much lower % having sub-cloud precipitation (very stringent). We changed the method to be more consistent with echoes designated at a height "90m" (3 gates) below the cloud base estimate, and specifying a threshold for the echoes to be at minimum -20 dBZ (or larger, additional sensitivity test needs discussed below). We also specify we were assuming a ceilometer for "cloud base" and echoes needing to be at least a few gates (90m) below that.

Here, "cloud base" is something the authors agree is not well-defined; the application is consistent with other ENA uses including those that we have referenced (e.g., Yang et al., 2018). We performed a quick sensitivity test for the Reviewer benefit related to how our %s change if the ceilometer-estimated cloud base was with different offsets, and in changing a minimum "Z" for that echo to be '-20dBZ' instead of '-30dBZ', etc. For the same -30dBZ threshold, the %s of drizzle echo at 30 m versus 90 m will understandably be higher: for the 400-600m thick cloud bin, 95% of them had an echo at 30 m below, whereas 77% had those echoes at 90 m below. Yet, if using

the '-20 dBZ' value instead, the % lowers from 95% to 25%, tied to the more significant drizzle threshold. Since our goal is something informative to significant drizzle presence, we opted to report the more conservative values.

Finally, we made alterations based on the ceilometer comment, interested in the references alluded to. We have added a reference in the revised manuscript from Zhu et al. (2024), which speaks to an application of high resolution lidar versus the ceilometer for cloud base (see Figure), and we assumed ceilometers as reasonable for "cloud base" to within 100 meters. But we agree several choices can lead to very different results.

Fig. 2. Schematic of the sampling strategy for the (a) ceilometer and (b) T2 lidar operated in the time-gated mode. The ceilometer produces a backscatter profile, while the T2 lidar only receives backscattered photons within the time-gated window. The T2 lidar scans the cloud-base region as shown in (b) and detailed in the text. (c) Backscatter profiles observed from ceilometer on 5 Jul 2022, at the CMAS site. The slanted solid black line indicates the scanning region of the T2 lidar. (d) A close-up view of the ceilometer-observed backscatter profile near cloud base observed at 2031:04 LT [shown as the yellow star in (c)]. (e) Profiles of backscattered photons observed by the T2 lidar in five gated windows near cloud base. Each colored profile indicates one second of observations from the T2 lidar in the time-gated window between 2031:16 and 2031:20 LT.

Zhu, Z., and Coauthors, 2024: Peering into Cloud Physics Using Ultra-Fine-Resolution Radar and Lidar Systems. Bull. Amer. Meteor. Soc., **105**, E2010–E2025, <a href="https://doi.org/10.1175/BAMS-D-23-0032.1">https://doi.org/10.1175/BAMS-D-23-0032.1</a>.

I 421-422 - To bring this point concerning drizzle home, I think that relative errors would be more meaningful and insightful, e.g., add to fig. 7 and 11 the same plots but for (LWP\_ref - LWP sphot)/LWP ref.

Figs. 7 and 11 have been modified according to the Reviewer suggestion for relative errors.

1430-431 - confusing sentence - recommend rewording.

**Modified.**

I 491 - "drizzle is present" --> "drizzle is likely present"

Ok.

I 493 (related to the second main comment) - remind the readers that we are discussing likely non-precipitating Sc clouds. We shouldn't forget that we are comparing conditioned datasets. By the same token, begin the item in I. 497 with "potential "Drizzle" signatures...

Ok.

Fig. 1: If not mentioned in the text prior to the first Fig. 1 reference, define KAZR here and provide a reference. Is this raw kazr data (if so, provide mode) or an advanced product such as ARSCL (provide a reference)? Also it is unclear from the caption what \*exactly\* the shaded regions represent and the text lacks that information as well.

Fixed several issues with the Figure 1 presentation and surrounding text.

Fig. 4 and elsewhere - recommend changing the color-bar title "observations" to "samples". Also, specify joint histogram bin widths.

Ok.

Fig. 7 and 11 - a curve showing the average diff vs. r\_e would be helpful in understanding this figure.

Ok. See comment above on revised Figs. 7 and 11.

Tables 1 and 2 and Tables 3 and 4 - I think that multi-panel figures merging the data in 1 and 2 (wind effects) and 3 and 4 (site differences) will scan much better with no additional analysis required.

Possible. Can inquire how the AMT technical editor / staffing might handle these changes.

Data availability statement - provide a reference for each data product. The ARM Data Discovery can generate those, to my knowledge.

Agree. Will add appropriate ARM / ADC DOI and data discovery references.

**Anonymous Referee #2**

The manuscript submitted by Sookdar et al. evaluates automated sun photometer retrieval of stratocumulus cloud optical depth, effective radius and liquid water path. The dataset consists of 6 years of data at ARM SGP in Oklahoma, USA and ARM ENA in the Azores, Portugal. For reference, they compared the sun photometer products (SPHOT) to products from Multifilter Rotating Shadowband Radiometer (MFRSR), Microwave Radiometer retrievals (MWRRET Versions 1 and 2), Tropospheric Optimal Estimate approach utilizing emitted radiance interferometer (TROPoe) and the baseline ARM retrieval of cloud microphysical properties from KARZ radar and microwave radiometer (MICROBASE).

In general the paper is well written in clear English language. It is a rather technical paper intercomparing different retrieval for the specific stratocumulus cloud conditions cast. It will certainly serve as a reference for studies utilizing these products. In addition, there is also a short analysis about representativeness of island stations which will be of interest by a larger audience. The paper is probably well suited in the AMT journal.

In the following, I write down some comments or remarks. In general I've recommended this manuscript as subject to minor revisions, although if the authors decide to take action as outlined in the comments, it might be a bit more work. We thank this Reviewer for their comments in attempting to make this manuscript suitable for publication in AMT. We hope we appropriately address the comments in the time allotted. We also note that several of our replies reference other comments to Reviewer 1.

• While the dataset description (L116 ff.) separates between in-cloud drizzle and virga, retrieval differences are later in the results and discussions mostly attributed to "drizzle". it is unclear wether this includes also the virga cases and should be clarified.

This question of sub-cloud precipitation, etc., seems to be a common concern of the Reviewers. We have attempted to clarify the text in these sections to better communicate what we intended as cloud base, how we determined if drizzle was possibly present, and other details. We think some confusion is also that we used sub-cloud "drizzle" and "virga" interchangeably, but this is not necessarily the same thing. Similarly, we may have water that accumulated on instruments as a result of precipitation that happened prior to the samples being collected. As also in our reply to Reviewer 1, we have attempted to clarify our data/use in the revised text.

 For the ENA site, the authors already demonstrate that the fraction of lower radar-derived cloud base height than those from the ceilometer can serve as a proxy for identifying drizzle or virga. This method could be used to filter out such cases entirely in the comparison. Here the authors could take the opportunity to strengthen the conclusion which accounts drizzle occurrence for retrieval differences.

We agree with Reviewer 2 that we could have adopted a more aggressive option towards avoiding cases with subcloud echo or significant in-cloud drizzle. As we responded to Reviewer 1, our intent was only to avoid negative influences of water accumulating on the instruments, but this may have been phrased in a manner suggesting we were attempting to avoid "drizzle" influence on all column/cloud measurements. We intended to include examples in columns with measurements influenced by the presence of drizzle. We were doing so since these are valuable for our community that has interest in the limits of these observations in the presence of drizzle.

• As stated in the text (L301), the MWRRET Version 2 incorporate the 89 Ghz channel to separate drizzle contribution to LWP. Could this capability not be used to identify or filter drizzle-affected cases at the SGP site as well?

We include the MWRRETv2 for SGP. It was an oversight that we did not include it originally for SGP, but an easy fix to add. From our understanding of how TROPoe was designed and implemented for ARM (communication with D. Turner), this code is different than MWRRETv2 in that the TROPoe does not use the 89 GHz channel; This was a choice as related to possible calibration issues with that channel for TROPoe purposes. TROPoe only uses the 23.8 and 30/31 GHz channels, though this choice does have an advantage of making its retrievals more consistent across different ARM locations (which may not all have MWR3C, etc., over time). We should note that it seems MWRRETv2 appears to perform much better (relatively) at SGP than ENA where drizzle / higher LWP is more common.

• In the datasets section the authors should elaborate on why MWRRET Version 1 is primarily used for comparisons, despite the availability of Version 2, which I guess is more advanced? There is a brief statement that Version 2 resemble TROPoe results but it is not shown or further discussed. I think it would improve transparency to include Version 2 results as well - could be also an Appendix figure.

As above, MWRRETv2 was used for ENA (the only option available), whereas MWRRETv1 and V2 were available for SGP. It was an oversight of the authors to not include V2 as well as V1. We agree that it is more useful to show readers MWRRETv2 as an option as well as TROPoe, as those retrievals are also different and may behave differently in the presence of possible drizzle.

In L205, the text refers to a constant reference cloud number concentration of 100 cm-3 for liquid water content calculation in Microbase. This likely should be "cloud particle or droplet number concentration (CDNC)"? Additionally, in L214, a different fixed value of 200 cm-3 is used for effective radius estimation. Why are two different assumptions used?

This is a similar question to one we addressed for Reviewer 1. We will repeat our answer from above for the benefit of this Reviewer as well:

The default MICROBASE product setting has not been modified by ARM since its initial production. Its key assumptions are not optimized for any one condition, but a "baseline" for a variety of SGP cloud conditions. This default setting has been reported as tuned using radiative closure studies. The details for these closure efforts do not appear to be well-documented in the literature discussing MICROBASE. Our assumption has been that tuning was done through the lens of the CDNC parameter, set to 200 cm-3. The product was never modified or "matched" to any other site, and ARM releases this product at other sites using the default SGP configuration.

For the LWC-Z mapping expression as from Liao and Sassen (1994): We believe its use is consistent within MICROBASE's approach on how to approximately partition LWP into discrete LWC intervals (height) by using radar reflectivity Z as a reference for where in this cloud one expects relative higher/lower LWC. This use is an assumed fit centered on a particular No (100 cm-3) condition. However, how this is applied in MICROBASE is arguably less important to its impact.

First, LWC  $\sim$  N\*D^3 and Z  $\sim$  N\*D^6, thus any change may introduce some discrepancies (i.e., shifts in how LWC might get placed) for the non-precipitating (smaller drop) times where the N control to LWC versus Z is more influential than D. But, the basic concept is that wherever there is a relatively large Z, it will be assigned a relatively large LWC. Yet, the total of all LWC is constrained by the LWP. Thus, this relationship does not create LWC in the column, it simply moves it around the cloud. The changes to Re are subtle (slightly higher in one place, slightly lower in another), and less significant to how Re retrievals (overall) are scaled by the primary CDNC control.

As Liao and Sassen (1994) note, [this expression is] "for estimating liquid water content only if the cloud droplet concentration No is known, [however] comparison with empirical relationships suggests that a value of No  $^{\sim}$  100 cm-3 produces satisfactory results in a variety of liquid phase clouds." In Liao and Sassen, the expression was intended for "direct" estimates, whereas for MICROBASE, what is important is how LWC profiles are partitioned in height. MICROBASE Re values are averaged in height and time, further smoothing the role of that LWC  $\rightarrow$  Re mapping.

Overall, there was not much value or control in modifying a "new" MICROBASE relationship specific to multiple No conditions, given the path constraint for LWC and the averaging we perform. The authors included a line in the revised manuscript to note this discrepancy.

• For comparison the Microbase CDNC are modified to 100 and 50 cm-3 and they argue that these values are more physical. This raises questions about using Microbase as the reference retrieval and needs stronger justification.

As in our previous response, this is a known challenge within the community given the lack of "baseline" references or standards for cloud property retrievals, and why we think it important to have additional standard/operational products for quantities such as Re, tau, and LWP. As we allude in replies to Reviewer 1, we suspect that part of MICROBASE's longevity (or enduring favorability) as a standard is that it is relatively simple in its assumptions, does not require radar instrument calibration, and it serves as a basic baseline for a variety of conditions. In that sense, its justification for inclusion is more that it is one of the few standard operational products of its kind, deficiencies and all. Nevertheless, this product was not matched to ENA, and we believe the CDNC modifications we performed (that aspect of its assumptions) are consistent with the previous MICROBASE tuning to yield some estimates better than "default" SGP settings.

---

## Author Response (AR2)

**Second Response to Reviewers:**

"Marine and Continental Stratocumulus Cloud Microphysical Properties Obtained from Routine ARM Cimel Sunphotometer Observations"

Sookdar et al. (2025) egusphere 2025-694

**General Comments from the Authors:**

We thank the Reviewers and the Editor for their continued efforts. For this response, we have again combined all replies into a single document. Author replies are in blue font.

**Anonymous Referee #1**

1. Final sentence of the abstract: The authors did reword that sentence, but I do not think it was toned down to reflect the limited extent of this island-effect analysis, and hence, the drawn conclusions. I believe that a simple change of "is" --> "could be" would do the trick, such that the sentence would read:

"Additional sensitivity tests for island influences on marine Sc properties suggest that while island-influenced winds may promote larger cloud LWP or thickness, the influence could be within retrieval method uncertainty and/or collocated instrument variability."

**Fixed.**

2. Relative errors: I appreciate the authors for implementing relative errors in the figures and discussion. I think that a sign change is required (in the figure and where relevant in the text) since errors are typically calculated as deviations of the tested method from the reference, so in the SPHOT TropOE case, it should be: SPHOT minus TropOE divided by TropOE instead of the current TropOE minus SPHOT divided by TropOE.

Good catch. In this case, it was the figure caption that was an author typo; the plot/relative error was correct (appropriate sign). We have fixed the text of the caption for Figure 7.

**Anonymous Referee #2**

- L119-L121: ".. since there was interest on the part of the authors .." - meaning is unclear. May rephrase the sentence to something like: "As LWP properties during sub cloud precipitation conditions are of high interest due to ..."

We have revised this sentence along the lines of the reviewer request.